# The origins and relatedness structure of mixed infections vary with local prevalence of *P. falciparum* malaria

Sha Joe Zhu[1†], Jason A Hendry[1†], Jacob Almagro-Garcia[1,2,3,4], Richard D Pearson[2,3,4], Roberto Amato[2,3,4], Alistair Miles[1,2,3,4], Daniel J Weiss[1], Tim CD Lucas[1], Michele Nguyen[1], Peter W Gething[1], Dominic Kwiatkowski[1,2,3,4], Gil McVean[1,3*], for the Pf3k Project

[1]Big Data Institute, Li Ka Shing Centre for Health Information and Discovery, University of Oxford, Oxford, United Kingdom; [2]Wellcome Centre for Human Genetics, University of Oxford, Oxford, United Kingdom; [3]Medical Research Council Centre for Genomics and Global Health, University of Oxford, Oxford, United Kingdom; [4]Wellcome Sanger Institute, Hinxton, United Kingdom

**\*For correspondence:**
gil.mcvean@bdi.ox.ac.uk

[†]These authors contributed equally to this work

**Competing interests:** The authors declare that no competing interests exist.

**Abstract** Individual malaria infections can carry multiple strains of *Plasmodium falciparum* with varying levels of relatedness. Yet, how local epidemiology affects the properties of such mixed infections remains unclear. Here, we develop an enhanced method for strain deconvolution from genome sequencing data, which estimates the number of strains, their proportions, identity-by-descent (IBD) profiles and individual haplotypes. Applying it to the Pf3k data set, we find that the rate of mixed infection varies from 29% to 63% across countries and that 51% of mixed infections involve more than two strains. Furthermore, we estimate that 47% of symptomatic dual infections contain sibling strains likely to have been co-transmitted from a single mosquito, and find evidence of mixed infections propagated over successive infection cycles. Finally, leveraging data from the Malaria Atlas Project, we find that prevalence correlates within Africa, but not Asia, with both the rate of mixed infection and the level of IBD.
DOI: https://doi.org/10.7554/eLife.40845.001

## Introduction

Individuals infected with malaria-causing parasites of the genus *Plasmodium* often carry multiple, distinct strains of the same species (*Bell et al., 2006*). Such mixed infections, also known as complex infections, are likely indicative of intense local exposure rates, being common in regions of Africa with high rates of prevalence (*Howes et al., 2016*). However, they have also been documented for *P. vivax* and other malaria-causing parasites (*Mueller et al., 2007*; *Collins, 2012*), even in regions of much lower prevalence (*Howes et al., 2016*; *Steenkeste et al., 2010*). Mixed infections have been associated with increased disease severity (*de Roode et al., 2005*) and also facilitate the generation of genomic diversity within the parasite, enabling co-transmission to the mosquito vector where sexual recombination occurs (*Mzilahowa et al., 2007*). The distribution of mixed infection duration, and whether the clearance of one or more strains results purely from host immunity (*Borrmann and Matuschewski, 2011*) or can be influenced by interactions between the distinct strains (*Enosse et al., 2006*; *Bushman et al., 2016*), are all open questions.

Although mixed infections can be studied from genetic barcodes (*Galinsky et al., 2015*), genome sequencing provides a more powerful approach for detecting mixed infections (*O'Brien et al., 2016*; *Chang et al., 2017*). Genetic differences between co-existing strains manifest as polymorphic loci in the DNA sequence of the isolate. The higher resolution of sequencing data allows the use of

statistical methods for estimating the number of distinct strains, their relative proportions, and genome sequences (*Zhu et al., 2018d*). Although genomic approaches cannot identify individuals infected multiple times by identical strains, and are affected by sequencing errors and problems of incomplete or erroneous reference assemblies, they provide a rich characterisation of within host diversity (*Manske et al., 2012*; *Auburn et al., 2012*; *Pearson et al., 2016*).

Previous research has highlighted that co-existing strains can be highly related (*Nair et al., 2014*; *Trevino et al., 2017*). For example, in *P. vivax*, 58% of mixed infections show long stretches of within host homozygosity (*Pearson et al., 2016*). In addition, (*Nkhoma et al., 2012*) reported an average of 78.7% *P. falciparum* allele sharing in Malawi and 87.6% sharing in Thailand. A mixed infection with related strains can arise through different mechanisms. Firstly, relatedness is created when distinct parasite strains undergo meiosis in a mosquito vector. A mosquito vector can acquire distinct strains by biting a single multiply-infected individual, or multiple infected individuals in close succession. Co-transmission of multiple meiotic progeny produces a mixed infection in a single-bite, containing related strains. Alternatively, relatedness in a mixed infection can result from multiple bites in a parasite population with low genetic diversity, such as is expected during the early stages of an outbreak or following severe population bottlenecks; for instance, those resulting from an intervention (*Mouzin et al., 2010*; *Wong et al., 2017*; *Daniels et al., 2015*). Interestingly, serial co-transmission of a mixed infection is akin to inbreeding, producing strains with relatedness levels well above those of standard siblings.

The rate and relatedness structure of mixed infections are therefore highly relevant for understanding regional epidemiology. However, progress towards utilising this source of information is limited by three problems. Firstly, while strain deconvolution within mixed infections has received substantial attention (*Galinsky et al., 2015*; *O'Brien et al., 2016*; *Chang et al., 2017*; *Zhu et al., 2018d*), currently, no methods perform both deconvolution of strains and estimation of relatedness. Because existing deconvolution methods assume equal relatedness along the genome, differences in relatedness that occur, for example through infection by sibling strains, can lead to errors in the estimation of the number, proportions and sequences of individual strains (*Figure 1*). Recently, progress has been made in the case of dual-infections with balanced proportions (*Henden et al., 2018*), but a general solution is lacking. The second problem is that little is known about how the rate and relatedness structure of mixed infections relates to underlying epidemiological parameters. Informally, mixed infections will occur when prevalence is high; an observation exploited by *Cerqueira et al. (2017)* when estimating changes in transmission over time. However, the quantitative nature of this relationship, the key parameters that influence mixed infection rates and how patterns of relatedness relate to infection dynamics are largely unexplored. Finally, an important issue, though not one addressed here, is the sampling design. Malaria parasites may be taken from individuals presenting with disease or as part of a surveillance programme. They are also often highly clustered in time and space. What impact different sampling approaches have on observed genomic variation is not clear. Nevertheless, because mixed infection rates are likely to respond rapidly to changes in prevalence (*Volkman et al., 2012*), exploring these challenges may render critical insights for malaria control in the field.

Here, we develop, test and apply an enhanced method for strain deconvolution called `DEploidIBD`, which builds on our previously-published `DEploid` software. The method separates estimation of strain number, proportions, and relatedness (specifically the identity-by-descent, or IBD, profile along the genome) from the problem of inferring genome sequences. This strategy provides substantial improvements to accuracy when strains are closely related. We apply `DEploidIBD` to 2344 field isolates of *P. falciparum* collected from 13 countries over a range of years (2001–2014) and available through the Pf3k Project (see Appendix), and characterise the rate and relatedness patterns of mixed infections. In addition, we develop a statistical framework for characterising the processes underlying mixed infections, estimating that nearly half of symptomatic mixed infections arise from the transmission of sibling strains, as well as demonstrating the propagation of mixed infections through multiple cycles of host-vector transmission. Finally, we investigate the relationships between statistics of mixed infection and epidemiological estimates of pathogen prevalence (*MAP, 2017*), showing that, at a global level, regional rates of mixed infection and levels of background IBD are correlated with estimates of malaria parasite prevalence.

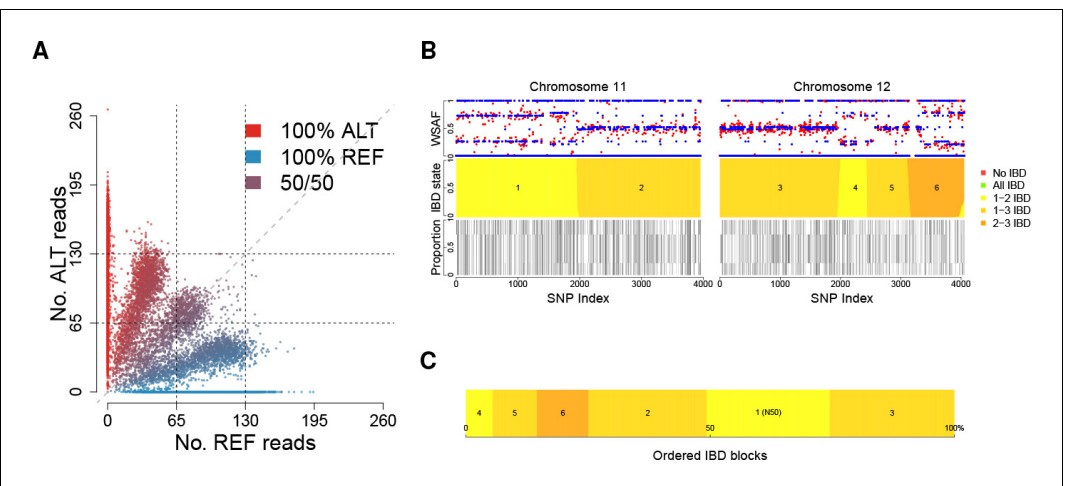

**Figure 1.** Deconvolution of a complex field sample PD0577-C from Thailand. (**A**) Scatter-plot showing the number of reads supporting the reference (REF: x-axis) and alternative (ALT: y-axis) alleles. The multiple clusters indicate the presence of multiple strains, but cannot distinguish the exact number or proportions. (**B**) The profile of within-sample allele frequency along chromosomes 11 and 12 (red points) suggests a changing profile of IBD with three distinct strains, estimated to be with proportions of 22%, 52% and 26% respectively (other chromosomes omitted for clarity, see *Figure 1—figure supplement 1*); blue points indicate expected allele frequencies within the isolate. However, the strains are inferred to be siblings of each other: green segments indicate where all three strains are IBD (Note: green segments do not appear in this example, but occur in Figure 5); yellow, orange and dark orange segments indicate the regions where one pair of strains are IBD but the others are not. In no region are all three strains inferred to be distinct. (**C**) Statistics of IBD tract length, in particular illustrating the N50 segment length. A graphical description of the modules and workflows for DEploidIBD is given in *Figure 1— figure supplement 2*.

DOI: https://doi.org/10.7554/eLife.40845.002

The following figure supplements are available for figure 1:

**Figure supplement 1.** Whole genome deconvolution of field sample PD0577-C.

DOI: https://doi.org/10.7554/eLife.40845.003

**Figure supplement 2.** A graphical overview of the data types and work flows for DEploidIBD.

DOI: https://doi.org/10.7554/eLife.40845.004

## Strain deconvolution in the presence of relatedness

Existing methods for deconvolution of mixed infections typically assume that the different genetic strains present in mixed infections are unrelated. This assumption allows for efficient computation of priors for allele frequencies within samples, either through assuming independence of loci (*O'Brien et al., 2016*) or as sequences generated as imperfect mosaics of some (predefined) reference panel (*Zhu et al., 2018d*). However, when strains are related to each other, and particularly when patterns of IBD vary along the genome (for example through being siblings), the constraints imposed on within-sample allele frequencies through IBD can cause problems for deconvolution methods, which can try to fit complex strain combinations (with relatedness) as simpler configurations (without relatedness). Below we outline the approach we take to integrating IBD into DEploid. Further details are provided in the Appendix.

### Decoding genomic relatedness among strains

A common approach to detecting IBD between two genomes is to employ a hidden Markov Model that transitions into and out of IBD states (*Chang et al., 2015*; *Gusev et al., 2009*; *Gusev et al., 2011*). We have generalised this approach to the case of $K$ haploid *Plasmodium* genomes (strains). In this setting, there are $2^K$ possible genotype configurations, as each of the $K$ strains can be either reference (i.e. same as the reference genome used during assembly), or alternative (i.e. carry a different allele) at a given locus (we assume all variation is bi-allelic). In most cases, if each of the $K$ strains constitutes a unique proportion of the infection, each genotype configuration will produce a distinct

alternative within sample allele frequency (WSAF; *Figure 1A*), which defines the expected fraction of total sequencing reads that are alternative at a given locus in the sequenced infection.

The effect of IBD among these $K$ strains is to limit the number of distinct genotype configurations possible, in a way that depends on the pattern of IBD sharing. Consider that, for any given locus, the $K$ strains in the infection are assigned to $j \leq K$ possible reference haplotypes. IBD exists when two or more strains are assigned to the same haplotype. In this scenario, the total number of possible patterns of IBD for a given $K$ is equal to $\sum_{j=1}^{K} S(K,j)$, where $S(K,j)$ is the number of ways $K$ objects can be split into $j$ subsets; a Stirling number of the second kind (*Graham et al., 1988*). Thus, for two strains, there are two possible IBD states (IBD or non-IBD), for three strains there are five states (all IBD, none IBD and the three pairwise IBD configurations), for four strains there are fifteen states (see Appendix), and so on. We limit analysis to a maximum of four strains for computational efficiency. Finally, for a given IBD state, only $2^j$ rather than $2^K$ genotype configurations are possible, thereby restricting the set of possible WSAF values.

Moving along the genome, recombination can result in changes in IBD state, hence changing the set of possible WSAF values at loci (*Figure 1B*). To infer IBD states we use a hidden Markov model, which assumes linkage equilibrium between variants for computational efficiency, with a Gamma-Poisson emission model for read counts to account for over-dispersion (see Appendix). Population-level allele frequencies are estimated from isolates obtained from a similar geographic region. Given the structure of the hidden Markov model, we can compute the likelihood of the strain proportions by integrating over all possible IBD sharing patterns, yielding a Bayesian estimate for the number and proportions of strains (see Appendix 1 Implementation details). We then use posterior decoding to infer the relatedness structure across the genome (*Figure 1B*). To quantify relatedness, we compute the mean IBD between pairs of strains and statistics of IBD tract length (mean, median and N50, the length-weighted median IBD tract length, *Figure 1C*).

In contrast to our previous work, `DEploidIBD` infers strain structure in two steps. In the first we estimate the number and proportions of strains using Markov Chain Monte-Carlo (MCMC), allowing for IBD as described above. In the second, we infer the individual genomes of the strains, using the MCMC methodology of , which can account for linkage disequilibrium (LD) between variants, but without updating strain proportions. The choice of reference samples for deconvolution is described in *Zhu et al. (2018d)* and in the Appendix. During this step we do not use the inferred IBD constraints per se, though the inferred haplotypes will typically copy from the same (or identical) members of the reference panel within the IBD tract.

## Results

### Method validation

#### Validation using experimentally generated mixed infections

We first sought to characterise the behaviour of `DEploidIBD` and compare its performance to the previously published method, `DEploid`. To this end, we re-analysed a set of 27 experimentally generated mixed infections (*Wendler, 2015*) that had been previously deconvoluted by `DEploid` (*Zhu et al., 2018d*) using `DEploidIBD` (*Figure 2—figure supplement 2*). These mixed infections were created with combinations of two or three laboratory strains (selected from 3D7, Dd2, HB3 and 7G8), set at varying known proportions (*Wendler, 2015*), and therefore provide a simple framework for evaluating inference of the number of strains ($K$) and their proportions. Since the method allows deconvolution of mixed infections containing up to four strains, we augmented the experimental mixtures by combining all four lab strains in silico at differing proportions (see Appendix 2 In silico lab mixtures). Using this approach, we found that `DEploid` and `DEploidIBD` performed comparably, except in the case of three strains with equal proportions, where LD information is necessary to achieve accurate deconvolution and `DEploid` performed better. Both `DEploid` and `DEploidIBD` struggled to deconvolute our in silico mixtures of four strains, typically underestimating the number of strains present.

## Validation against simulated mixed infections

Validation using mixtures of lab strains has two limitations: (i) the strains comprising the mixed infection were part of the reference panel and (ii) no IBD was present. We therefore investigated the ability of `DEploidIBD` to recover IBD between strains within a mixed infection, in the context of a realistic reference panel, and with strains typical of those we observe in nature. To achieve this, we designed a validation framework where clonal samples from the Pf3k project were combined in silico to produce simulated mixed infections, allowing us to create examples with varying numbers of strains and proportions, and to introduce tracts of IBD, by copying selected sections of the genome between strains. Using this framework, we constructed a broad suite of simulated mixed infections, derived from clonal samples from Africa and Asia that were combined into mixtures of 2, 3 and 4 strains with variable proportions and IBD configurations.

We randomly selected 189 clonal samples of African origin and 204 clonal samples of Asian origin from which to construct our simulated mixed infections and restricted the analysis to chromosome 14 to reduce computational time. Starting with mixed infections containing two strains, we randomly took two samples of African or Asian origin and combined them at proportions ranging from highly imbalanced (10% and 90%) to exactly balanced (each 50%) and used copying to produce either no (0%), low (25%), medium (50%) or high (75%) levels of IBD (note that background IBD between the two clonal strains may also exist). In total this resulted in 4,000 $K = 2$ mixed infections, each of which was deconvoluted with `DEploid` and `DEploidIBD`. Outputs of `DEploidIBD` were compared to the true values for each simulated infection, including the inference of $K$, the effective $K_e$ (computed as $K_e = 1/\sum w_i^2$, where $w_i$ is the proportion of the $i$th strain, thus incorporating proportion inference), the average pairwise relatedness between strains (for $K = 2$, this is the fraction of the genome inferred to be IBD), and the inference of IBD tract length, expressed as the IBD N50 metric.

For mixtures of two strains, both `DEploid` and `DEploidIBD` performed well in scenarios where the IBD between strains was low (<=25%). In moderate or high IBD scenarios with imbalanced strain proportions, `DEploid` tended to underestimate the proportion of the minor strain resulting in underestimation of $K_e$, whereas `DEploidIBD` was able to infer the proportion of these mixtures correctly (*Figure 2*). The main novelty of `DEploidIBD` is the calculation of an IBD profile between strains. We found that the IBD summary statistics produced by `DEploidIBD` were accurate across all two-strain mixed infections tested in Africa. In Asia, `DEploidIBD` tended to estimate more IBD than was simulated (*Figure 2B*). However, this likely reflects the presence of higher background IBD in Asia rather than systematic error.

To simulate realistic mixed infections containing 3 or 4 strains, we first considered the different transmission scenarios under which they can arise. We modelled a mixed infection of $K$ strains as resulting from $b$ biting events, where $K \in \{3, 4\}$ and $1 \leq b \leq K$. When greater than one strain is transmitted in a single biting event, the co-transmitted strains will share IBD, as a consequence of meiosis occurring in the mosquito. Strains transmitted through independent bites, causing superinfection in the host, do not share any IBD beyond background. Following this paradigm, we generated a suite of mixed infection types: $K = 3$, $b = 1, 2, 3$ and $K = 4$, $b = 1, 2, 2, 3, 4$ (the first $b = 2$ has two strains per bite, the second three and one); and simulated each of these across a variety of proportions, again using sets of clonal samples from Africa and Asia as starting strains.

As with the experimental validation, the balanced-proportion $K = 3$ mixed infections generated in silico proved challenging to deconvolute, with both methods inferring the presence of two rather than three strains (*Figure 2C*). In mixed infections with imbalanced proportions, we found that, in African samples with IBD ($b = 1, 2$), `DEploid` tended to either underestimate the number of strains present, or infer proportions incorrectly. In Asian samples this is less of an issue as the reference panels can provide better prior information for deconvolution due to lower diversity. In contrast, `DEploidIBD` consistently gave the correct number of strains and proportions in such cases, and produced IBD statistics that were accurate as long as the median coverage of simulated infections was > 20x. Both methods struggled to deconvolute mixed infections of four strains (*Figure 2—figure supplement 2*), although performed better (i.e. inferred $K = 4$ greater than 50% of the time) for mixtures with less IBD ($b = 3, 4$). However, even in these cases, estimates of the proportions and IBD statistics were variable, indicating that further work is needed before $K = 4$ mixed infections can be reliably deconvoluted.

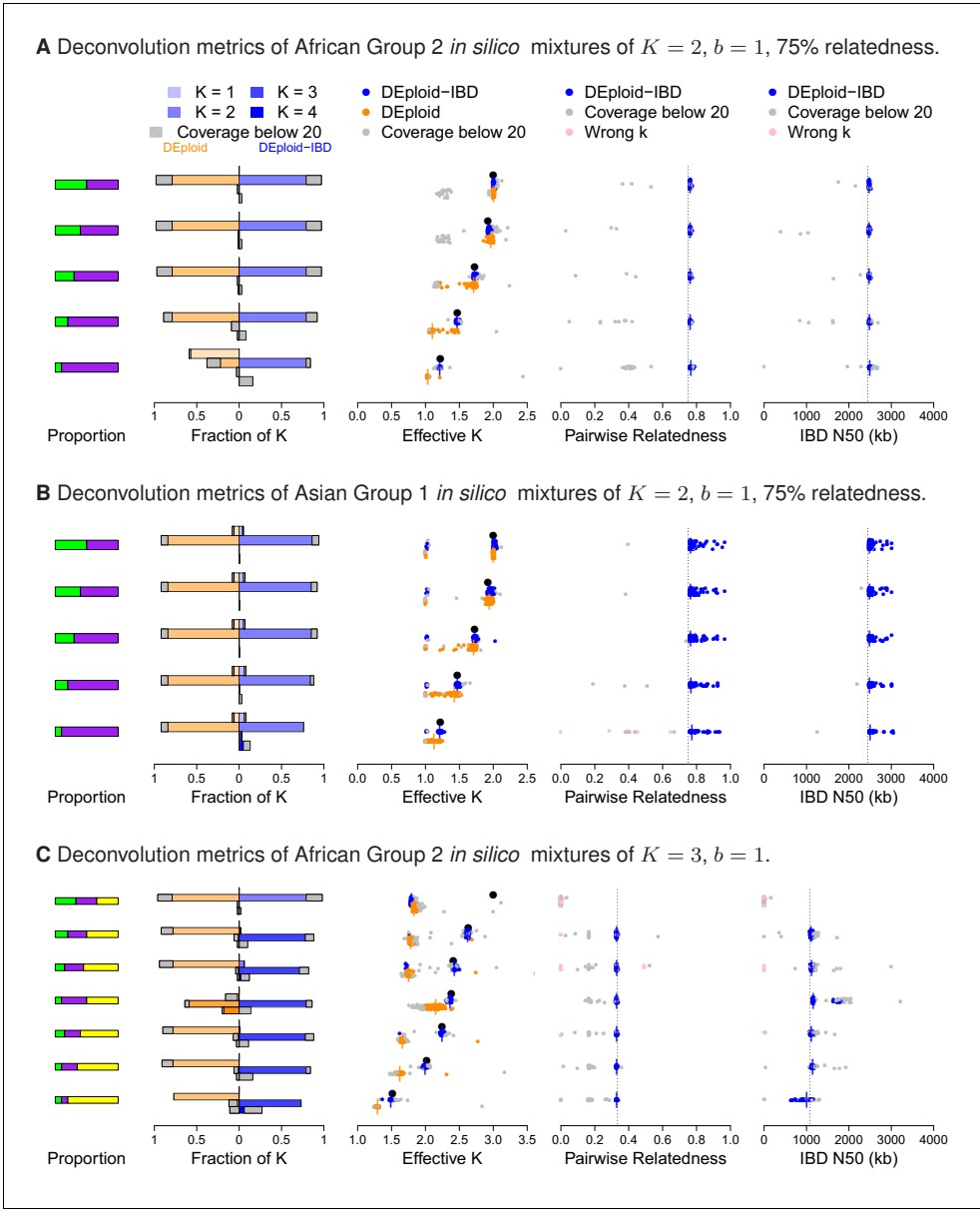

**Figure 2.** Performance of `DEploidIBD` and `DEploid` on 100 in silico mixtures for each of three different scenarios. From the left to the right, the panels show the strain proportion compositions, distribution of inferred *K* in a vertically-oriented histogram (top: *K* = 1, bottom: *K* = 4), using both methods: `DEploid` in orange and `DEploidIBD` in blue, effective number of strains, pairwise relatedness and IBD N50 (the latter two only for `DEploidIBD`). From top to the bottom, cases are ordered from even strain proportions to the most imbalanced composition. Grey points identify experiments of low coverage data (median sequencing depth < 20), and pink identify cases where *K* is inferred incorrectly. (**A**) In silico mixtures of two African strains with high-relatedness (75%) for 7757 (s.d. 178) sites on Chromosome 14, Note that `DEploid` underestimates the minor strain proportion if strains have high relatedness. In the extreme case, `DEploid` misclassifies a *K* = 2-mixture as clonal, whereas `DEploidIBD` consistently estimates the correct proportions. (**B**) In silico mixtures of two Asian strains with high-relatedness (75%) for 3041 sites (s.d. 227) on Chromosome 14, Note that `DEploid` underestimates strain number when the minor strain is low frequency, while `DEploidIBD` typically performs well. (**C**) In silico mixtures of three African strains, where each pair is IBD over a distinct third of the chromosome. Note that both methods fail to deconvolute the case of equal proportions. However, for unbalanced mixtures, `DEploidIBD` consistently performs better than `DEploid`.

DOI: https://doi.org/10.7554/eLife.40845.005

The following figure supplements are available for figure 2:

*Figure 2 continued on next page*

*Figure 2 continued*

**Figure supplement 1.** Validation of `DEploidIBD` using 27 in vitro lab mixtures and four in silico mixtures.
DOI: https://doi.org/10.7554/eLife.40845.006
**Figure supplement 2.** Illustration of simulation study design.
DOI: https://doi.org/10.7554/eLife.40845.007
**Figure supplement 3.** Additional comparison of `DEploidIBD` and `DEploid` on 100 in silico mixtures of two strains from Africa with low and moderate relatedness, illustrated by sub panels (**A**) and (**B**), respectively.
DOI: https://doi.org/10.7554/eLife.40845.008
**Figure supplement 4.** Additional comparison of `DEploidIBD` and `DEploid` on in silico $b = 2$ bite mixtures of $K = 3$ strains from Africa and Asia, illustrated by sub panels (**A**) and (**B**), respectively.
DOI: https://doi.org/10.7554/eLife.40845.009
**Figure supplement 5.** Comparison of `DEploidIBD` and `DEploid` on 100 in silico $b = 3$ bite mixtures of four strains from Africa.
DOI: https://doi.org/10.7554/eLife.40845.010

---

Finally, we used the in silico approach to explore the quality of haplotypes inferred by `DEploidIBD`, focusing on $K = 2$ infections across variable proportions. We compared the haplotype inferences between `DEploid` and DEpoloidIBD using the error model described in the Appendix, and found that rates of genotype error are similar for the two approaches in settings of low relatedness (`DEploidIBD` has an error rate of 0.7% per site per strain for 20/80 mixtures and 1.4% for 50/50 mixtures). However, for the 20/80% mixtures with high relatedness, genotype error for `DEploid` increased to 1.8%, while remaining at 0.8% for `DEploidIBD` (*Figure 3A*). Switch errors in haplotype estimation are comparable between the two methods and decrease with increasing relatedness due to higher homozygosity (*Figure 3B*). Finally, we identified a simple metric to compute on inferred haplotypes that can identify low quality haplotypes (see Appendix).

## Geographical variation in mixed infection rates and relatedness

To investigate how the rate and relatedness structure of mixed infections varies among geographical regions with different epidemiological characteristics, we applied `DEploidIBD` to 2344 field samples of *P. falciparum* released by the Pf3k project (*Pf3k Consortium, 2016*). These samples were collected under a wide range of studies with differing designs, though the majority of samples were collected from symptomatic individuals seeking clinical treatment. An important exception are the samples from Senegal which, though collected passively at a clinic, were screened to contain only one strain by SNP barcode (*Daniels et al., 2015*). A summary of the data sources is presented in *Table 1* and full details regarding study designs can be found at https://www.malariagen.net/projects/pf3k#sampling-locations. Details of data processing are given in the Appendix. For deconvolution, samples were grouped into geographical regions by genetic similarity; four in Africa, and three in Asia. (*Table 1*). Reference panels were constructed from the clonal samples found in each region. Since previous research has uncovered strong population structure in Cambodia (*Miotto et al., 2013*), we stratified samples into West and North Cambodia when performing analysis at the country level. Diagnostic plots for the deconvolution of all samples can be found at https://github.com/mcveanlab/mixedIBD-Supplement (*Zhu, 2018a*; copy archived at https://github.com/elifesciences-publications/mixedIBD-Supplement) and inferred haplotypes can be accessed at ftp://ngs.sanger.ac.uk/production/pf3k/technical_working/release_5/mixedIBD_paper_haplotypes/. We identified 787 samples where low sequencing coverage or the presence of low-frequency strains resulted in unusual haplotypes (see Appendix). Estimates of strain number, proportions and IBD states from these samples are used in subsequent analyses, but not the haplotypes. We also confirmed that reported results are not affected by the exclusion of samples with haplotypes with low confidence (data not shown). In all following analyses, only strains present with a proportion of 1% in a sample are reported.

We find substantial variation in the rate and relatedness structure of mixed infections across continents and countries. Within Africa, rates of mixed infection vary from 29% in The Gambia to 63% in Malawi (*Figure 4A*). Senegal has a rate of mixed infection (18%) lower than The Gambia, however as these samples were screened by SNP barcode to be clonal, this rate should be an underestimate. In Southeast Asian samples, mixed infection rates are in general lower, though also vary considerably;

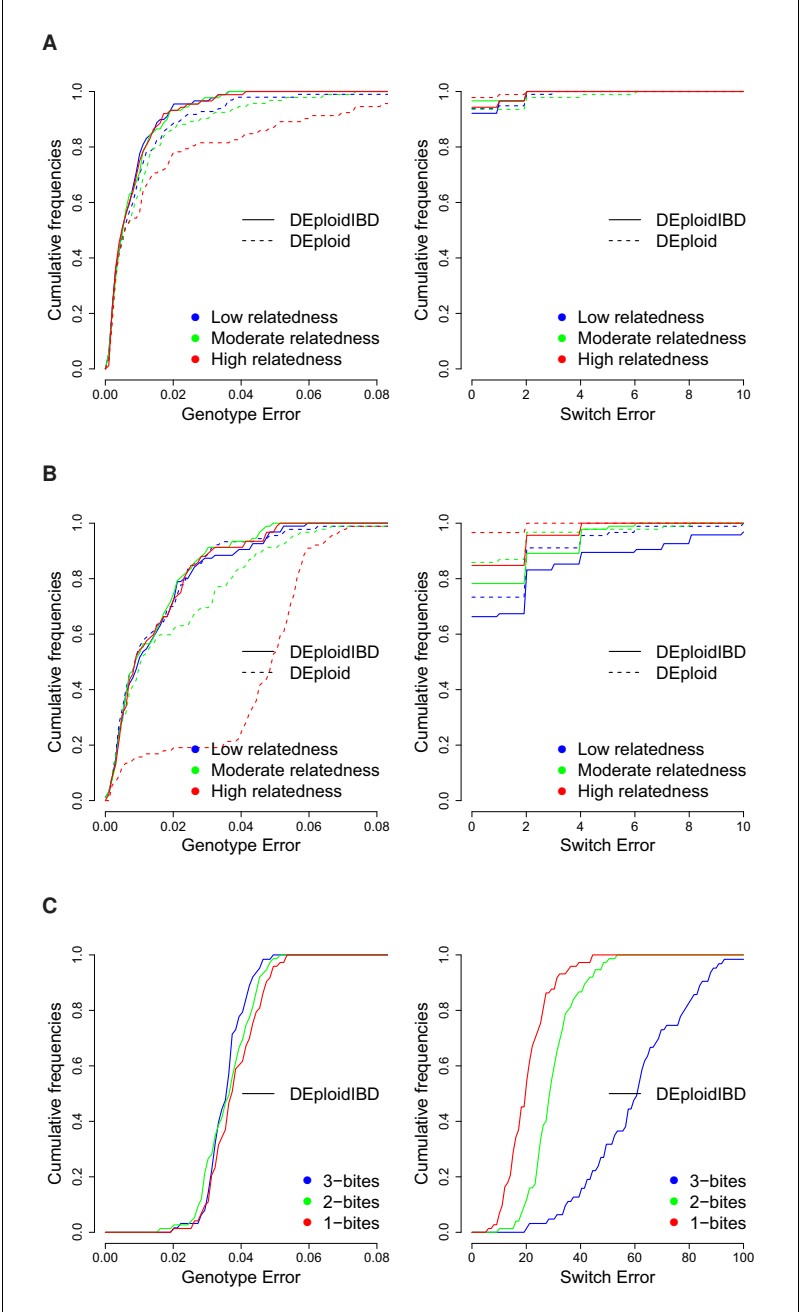

**Figure 3.** Cumulative distribution of the average per site genotype error (left) and switch error (right) across simulated mixtures (measured at sites that are heterozygous in the sample or sample-specific reference panel). (**A**) Error rates of Asian in silico samples of three levels of IBD (25%, 50% and 75%) for a $K = 2$ mixture with proportions of 20/80%. Because `DEploidIBD` estimates proportions more accurately, it enables better haplotype inference. (**B**) Error rates of African in silico samples of three levels of IBD (25%, 50% and 75%) for a $K = 2$ mixture with proportions of 20/80%. Inference in Asia benefits from better reference panels (due to lower overall diversity) and therefore gives lower error rates than in Africa. (**C**) `DEploidIBD` error rates for African in silico samples of three mosquito biting scenarios for a $K = 3$ mixture with proportions of 10/10/80%. The additional strain increases the difficulty of haplotype inference, particularly in the case of three independent bites.
DOI: https://doi.org/10.7554/eLife.40845.011

**Table 1.** Summary of Pf3k samples in data release 5.1, where $\bar{D}$ denotes mean read depth and $ss$ is sample size. Genotyping, including both indel and SNP variants, was performed using a pipeline based on GATK best practices, see Materials and methods. Data available from ftp://ngs.sanger.ac.uk/production/pf3k/release_5/5.1. *PfPR* is the inferred parasite prevalence rate in a 5 × 5 km resolution grid from the MAP project, centred at the Pf3k sample collection sites; Relatedness $\rho$ and effective number of strains $K_e$ are summary metrics from `DEploidIBD` output.

| Country | Year | Location | PfPR | ss | $\bar{D}$ (s.e.) | $\bar{\rho}$ | $\bar{K}_e$ | Reference |
|---|---|---|---|---|---|---|---|---|
| Gambia | 2008 | Brikam | 0.06 | 65 | 129 ( 9.4 ) | 0.5 | 1.3 | (*Amambua-Ngwa et al., 2012*) |
| Ghana | 2009 | Navrongo | 0.79 | 121 | 86 ( 5.7 ) | 0.21 | 1.6 | (*Duffy et al., 2015*; *Kamau et al., 2015*; *MalariaGEN Plasmodium falciparum Community Project, 2016*) |
| | 2010 | Navrongo | 0.79 | 171 | 127 ( 10.3 ) | 0.23 | 1.5 | |
| | 2011 | Navrongo | 0.72 | 97 | 76 ( 5.3 ) | 0.21 | 1.5 | |
| | | Kintampo | 0.58 | 6 | 89 ( 13.5 ) | 0.11 | 1.5 | |
| | 2012 | Navrongo | 0.52 | 47 | 111 ( 3.8 ) | 0.29 | 1.6 | |
| | | Kintampo | 0.41 | 40 | 157 ( 8.1 ) | 0.22 | 1.6 | |
| | 2013 | Navrongo | 0.31 | 88 | 119 ( 4 ) | 0.26 | 1.6 | |
| | | Kintampo | 0.29 | 4 | 172 ( 38.4 ) | 0.44 | 1.1 | |
| Malawi | 2011 | Chikwawa | 0.19 | 230 | 101 ( 3 ) | 0.26 | 1.7 | (*Ocholla et al., 2014*) |
| | | Zomba | 0.34 | 35 | 89 ( 9.1 ) | 0.24 | 1.6 | |
| Mali | 2007 | Bandiagara | 0.43 | 9 | 95 ( 25.2 ) | 0.39 | 1.8 | (*Mobegi et al., 2014*; *MalariaGEN Plasmodium falciparum Community Project, 2016*) |
| | | Faladje | 0.37 | 36 | 75 ( 10.1 ) | 0.27 | 1.3 | |
| | | Kolle | 0.21 | 51 | 82 ( 10.5 ) | 0.3 | 1.6 | |
| Guinea | 2011 | Nzerekore | 0.49 | 97 | 77 ( 4.6 ) | 0.17 | 1.4 | |
| Congo DR | 2013 | Kinshasa | 0.24 | 113 | 49 ( 3.2 ) | 0.31 | 1.5 | |
| Senegal | 2004 | Thies | 0.09 | 2 | 130 ( 68.2 ) | 0.01 | 1.4 | (*Wong et al., 2017*) |
| | 2009 | Thies | 0.04 | 43 | 175 ( 14.9 ) | 0.43 | 1.1 | |
| | 2010 | Thies | 0.04 | 24 | 159 ( 9.7 ) | 0.3 | 1.3 | |
| | 2011 | Thies | 0.03 | 32 | 97 ( 6 ) | 0.33 | 1.1 | |
| West Cambodia | 2009 | Pursat | 0.0071 | 19 | 75 ( 8.8 ) | 0.39 | 1.3 | (*Amato et al., 2017*; *MalariaGEN Plasmodium falciparum Community Project, 2016*) |
| | 2010 | Pursat | 0.0071 | 105 | 95 ( 6.8 ) | 0.65 | 1.2 | |
| | 2011 | Pailin | 0.0025 | 49 | 54 ( 4.1 ) | 0.43 | 1.1 | |
| | | Pursat | 0.0096 | 103 | 49 ( 3.1 ) | 0.63 | 1.2 | |
| | 2012 | Pailin | 0.00096 | 31 | 46 ( 5.6 ) | 0.43 | 1.0 | |
| | | Pursat | 0.0079 | 7 | 37 ( 19.1 ) | 0.58 | 1.4 | |
| North Cambodia | 2010 | Ratanakiri | 0.0039 | 50 | 71 ( 6.1 ) | 0.43 | 1.3 | |
| | 2011 | Preah Vihear | 0.02 | 73 | 51 ( 5.3 ) | 0.36 | 1.2 | |
| | | Ratanakiri | 0.0032 | 81 | 45 ( 4.3 ) | 0.47 | 1.4 | |
| | 2012 | Preah Vihear | 0.0075 | 30 | 43 ( 6.7 ) | 0.37 | 1.0 | |
| | | Ratanakiri | 0.0016 | 15 | 44 ( 8.9 ) | 0.3 | 1.3 | |
| Thailand | 2011 | Mae Sot | 0.00011 | 35 | 66 ( 7.5 ) | 0.35 | 1.2 | (*Miotto et al., 2013*; *MalariaGEN Plasmodium falciparum Community Project, 2016*) |
| | | Sisakhet | 1e-04 | 5 | 112 ( 25.4 ) | 0.17 | 1.3 | |
| | 2012 | Mae Sot | 5.7e-05 | 69 | 83 ( 4.9 ) | 0.58 | 1.3 | |
| | | Ranong | 0.00018 | 11 | 82 ( 12.4 ) | 0.38 | 1.2 | |
| | | Sisakhet | 0 | 13 | 89 ( 13 ) | 0.37 | 1.1 | |
| | 2013 | Sisakhet | 0 | 3 | 62 ( 8.8 ) | 0.09 | 1.2 | |
| Bangladesh | 2012 | Ramu | 0.0021 | 50 | 53 ( 4.2 ) | 0.45 | 1.5 | |

*Table 1 continued on next page*

*Table 1 continued*

| Country | Year | Location | PfPR | ss | $\bar{D}$ (s.e.) | $\bar{\rho}$ | $\bar{K}_e$ | Reference |
|---------|------|----------|------|-----|-----------------|-------------|-------------|-----------|
| Viet Nam | 2011 | Bu Gia Map | 0.0073 | 43 | 67 ( 5 ) | 0.43 | 1.3 | |
| | | Phuoc Long | 0.0053 | 27 | 68 ( 7.2 ) | 0.37 | 1.2 | |
| | 2012 | Bu Gia Map | 0.0072 | 19 | 115 ( 8 ) | 0.67 | 1.1 | |
| | | Phuoc Long | 0.0048 | 5 | 107 ( 6.3 ) | 0.81 | 1.2 | |
| Myanmar | 2011 | Bago Division | 0.0076 | 12 | 59 ( 7.1 ) | 0.24 | 1.2 | |
| | 2012 | Bago Division | 0.0084 | 47 | 62 ( 5.2 ) | 0.45 | 1.2 | |
| Laos | 2011 | Attapeu | 0.0094 | 59 | 71 ( 4.2 ) | 0.36 | 1.4 | |
| | 2012 | Attapeu | 0.02 | 25 | 77 ( 7.2 ) | 0.68 | 1.3 | |

DOI: https://doi.org/10.7554/eLife.40845.012

from 21% in Thailand to 54% in Bangladesh. Where data for a location is available over multiple years, we find no evidence for significant fluctuation over time (though we note that these studies are typically not well powered to see temporal variations). We observe that between 5.1% (Senegal) and 40% (Malawi) of individuals have infections carrying more than two strains.

Relatedness between samples and populations also varies substantially. In dual infections, the average fraction of the genome inferred to be IBD ranges from 14% in Guinea to 65% in West Cambodia (*Figure 4B*). Asian populations show, on average, a higher level of relatedness within dual infections (44%) compared to African populations ( 26%). Levels of IBD in samples with three or more strains are comparable to those seen in dual infections (average IBD being 45% in Asia and 37% in Africa) and significantly correlated at the country level, with correlation of 0.75 (p=0.0019,

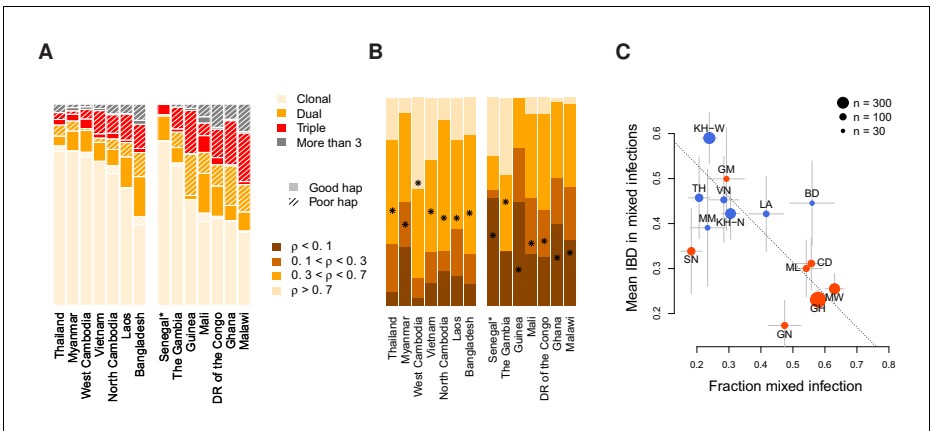

**Figure 4.** Characterisation of mixed infections across 2344 field samples of *Plasmodium falciparum*. (**A**) The fraction of samples, by population, inferred by `DEploidIBD` to be $K = 1$ (clonal), $K = 2$ (dual), $K = 3$ (triple), or $K = 4$ (More than 3). Populations are ordered by rate of mixed infections within each continent. We use shaded regions to indicate the distribution of 787 samples that have low-confidence deconvoluted haplotypes. Senegal is marked with an asterisks as these samples were screened to be clonal. (**B**) The distribution of average pairwise IBD sharing within mixed infections (including dual, triple and quad infections), broken down into unrelated (where the fraction of the genome inferred to be IBD, $\rho$, is <0.1), low IBD (0.1 $\geq \rho$<0.3), sib-level (0.3 $\geq \rho$<0.7) and high ($\rho \geq 0.7$). Stars indicate the average IBD scaled between 0 and 1 from bottom to the top. Populations follow the same order as in Panel A. (**C**) The relationship between the rate of mixed infection and level of IBD. Populations are coloured by continent, with size reflecting sample size and error bars showing ±1 s.e.m.. The dotted line shows the slope of the regression from a linear model. Abbreviations: SN-Senegal, GM-The Gambia, NG-Nigeria, GN-Guinea, CD-The Democratic Republic of Congo, ML-Mali, GH-Ghana, MW-Malawi, MM-Myanmar, TH-Thailand, VN-Vietnam, KH-Cambodia, LA-Laos, BD-Bangladesh.

DOI: https://doi.org/10.7554/eLife.40845.014

weighted by the number of mixed samples). Overall, 51% of all mixed infections involve strains with over 30% of the genome being IBD.

We next considered the relationship between mixed infection rate and the level of IBD. We find that populations with higher rates of mixed infection tend to have lower levels of IBD within mixed infections (linear model p=0.06 after accounting for a continental level difference and weighted by sample size). However, the continental level effect is driven by Senegal, which has an unusual combination of low mixed infections and also low IBD. Excluding Senegal, we find a consistent pattern across populations (*Figure 4C*), with a strong negative correlation between mixed infection rate and the level of IBD (Pearson $r = -0.65$, p = $3 \times 10^{-4}$). Previous work has demonstrated how a recent and dramatic decline in *P. falciparum* prevalence within Senegal has left an impact on patterns of genetic variation (*Daniels et al., 2015*), which may explain its unusual profile.

## Inferring the origin of IBD in mixed infections

The high levels of IBD observed in many mixed infections suggest the presence of sibling strains (*Figure 5*). To quantify the expected IBD patterns between siblings, we developed a meiosis simulator for *P. falciparum* (`pf-meiosis`), incorporating relevant features of malaria biology that can impact the way IBD is produced in a mosquito and detected in a human host. Most importantly, a single infected mosquito can undergo multiple meioses in parallel, one occurring for each oocyst that forms on the mosquito midgut (*Ghosh et al., 2000*). In a mosquito infected with two distinct strains, each oocyst can either self (the maternal and paternal strain are the same) or outbreed (the maternal and paternal strains are different). We model a $K = n$ mixed infection as a sample of $n$ strains (without replacement, as drawing identical strains yields $K = n - 1$) from the pool of strains created by all oocysts. Studies of wild-caught *Anopheles Gambiae* suggest that the distribution of oocysts is roughly geometric, with the majority of infected mosquitoes carrying only one oocyst

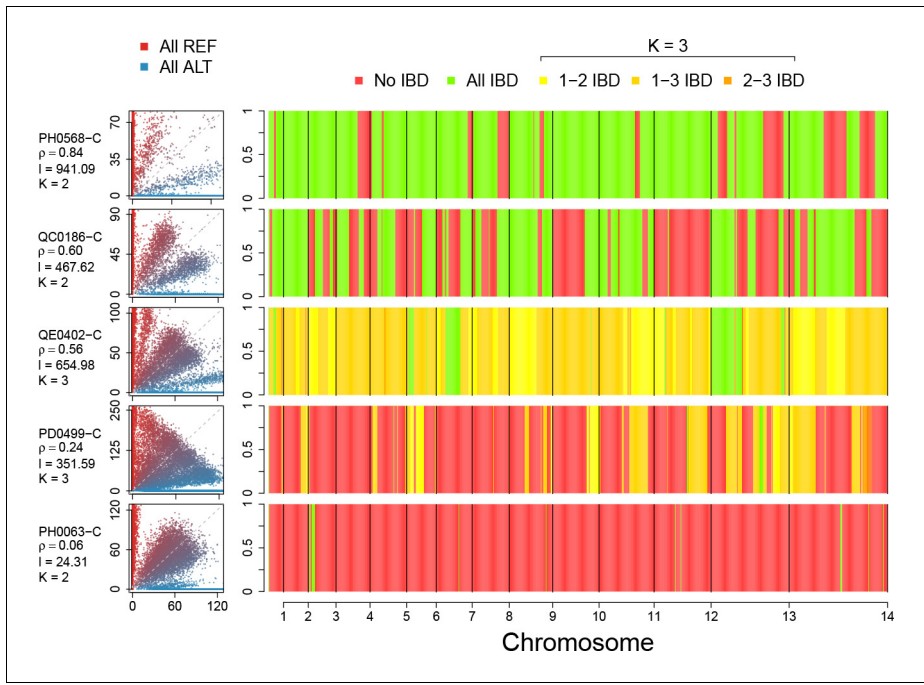

**Figure 5.** Example IBD profiles in mixed infections. Plots showing the ALT versus REF plots (left hand side) and inferred IBD profiles along the genome for five strains of differing composition. From top to bottom: A dual infection of highly related strains ($\rho = 0.84$); a dual infection of two sibling strains ($\rho = 0.6$); a triple infection of three sibling strains (note the absence of stretches without IBD); a triple infection of two related strains and one unrelated strain; and a triple infection of three unrelated strains. The numbers below the sample IDs indicate the average pairwise IBD, *r*, the mean length of IBD segments, *I*, in kb and the inferred number of distinct strains, *K*, respectively.

DOI: https://doi.org/10.7554/eLife.40845.015

(*Beier et al., 1991*; *Collins et al., 1984*). In such a case, we find that a $K = 2$ infection will have an expected IBD of 1/3, consistent with the observations of *Wong et al. (2018)*. Conditioning on at least one progeny originating from an outbred oocyst (such that a detectable recombination event has occurred), the expected IBD asymptotically approaches 1/2 as the total number of oocysts grows (see Appendix).

Using this simulation framework, we sought to classify observed mixed infections based on their patterns of IBD. We used two summary statistics to perform the classification: mean IBD segment length and IBD fraction. We built empirical distributions for these two statistics for each country in Pf3k, by simulating meiosis between pairs of clonal samples from that country. In this way, we control for variation in genetic diversity (as background IBD between clonal samples) in each country. Starting from a pair of clonal samples ($M = 0$, where $M$ indicates the number of meioses that have occurred), we simulated three successive rounds of meiosis ($M = 1, 2, 3$), representing the creation and serial transmission of a mixed infection (*Figure 6A*). Each round of meiosis increases the amount of observed IBD. For example, in Ghana, the mean IBD fraction for $M = 0$ was 0.002, for $M = 1$ was 0.41, for $M = 2$ was 0.66, and for $M = 3$ was 0.80 (*Figure 6B*). West Cambodia, which has lower genetic diversity, had a mean IBD fraction of 0.08 for $M = 0$ and consequently, the mean IBD fractions for higher values of $M$ were slightly increased, to 0.46, 0.68, 0.81 for $M = 1, 2$ and $3$, respectively (*Figure 6B*).

With these simulated distributions, we used Naive Bayes to classify $K = 2$ mixed infections in Pf3k (*Figure 6C*). Of the 393 $K = 2$ samples containing only high-quality haplotypes (see Appendix), 325 (83%) had IBD statistics that fell within the range observed across all simulated $M$. Of these, nearly half (183, 47%) were classified as siblings ($M > 0$). Moreover, we observe geographical differences in the rate at which sibling and unrelated mixed infections occur. Notably, in Asia a greater fraction of all mixed infections contained siblings (59% vs. 41% in Africa), driven by a higher frequency of $M = 2$ and $M = 3$ mixed infections (*Figure 6D*). Mixed infections classified as $M > 1$ are produced by serial co-transmission of parasite strains, that is a chain of mixed infections along which IBD increases.

## Characteristics of mixed infections correlate with local parasite prevalence

To assess how characteristics of mixed infections relate to local infection intensity, we obtained estimates of *P. falciparum* prevalence (standardised as $PfPR_{2-10}$, prevalence in the 2-to-10 year age range) from the Malaria Atlas Project ((*MAP, 2017*), see *Table 1*). The country-level prevalence estimates range from 0.01% in Thailand to 55% in Ghana, with African countries having up to two orders of magnitude greater values than Asian ones (mean of 36% in Africa and 0.6% in Asia). However, seasonal and geographic fluctuations in prevalence mean that, conditional on sampling an individual with malaria, local prevalence may be much higher than the longer-term (and more geographically widespread) country-level average, hence we extracted the individual pixel-level estimate of prevalence (corresponding to a 5 km × 5 km region) from MAP nearest to each genome collection point. We summarise mixed infection rates by the average effective number of strains, which reflects both the number and proportion of strains present.

Given that samples from Senegal were screened to be primarily single-genome (*Daniels et al., 2015*), we computed all correlations with prevalence including ($r_{S+}$) and excluding them ($r_{S-}$; *Figure 7*). We find that the effective number of strains is a significant predictor of $PfPR_{2-10}$ globally ($r_{S+} = 0.65, p < 10^{-5}$) and in African populations when Senegal is included ($r_{S+} = 0.48, p = 0.04$, $r_{S-} = 0.18, p = 0.51$), but is uncorrelated across Asia. Similarly, $PfPR_{2-10}$ is negatively correlated with background IBD globally ($r_{S+} = -0.43, p = 0.004$) and across Africa but not in Asia. Surprisingly, the amount of IBD observed within $K = 2$ mixed infections was not correlated with prevalence in Africa or Asia. The rate of sibling infection ($M = 1$) is not correlated with the parasite prevalence (Asia: $r_{S+} = 0.23, p = 0.2$, Africa: $r_{S+} = 0.16, p = 0.5$). However, the rate of supersiblings ($K = 2, M > 1$) is significantly correlated with $PfPR_{2-10}$ ($r_{S+} = -0.31, p = 0.04$) at the global scale, suggesting that serial co-transmission may occur more readily in low prevalence regions.

## Discussion

It has long been appreciated that mixed infections are an integral part of malaria biology. However, determining the number, proportions, and haplotypes of the strains that comprise them has proven

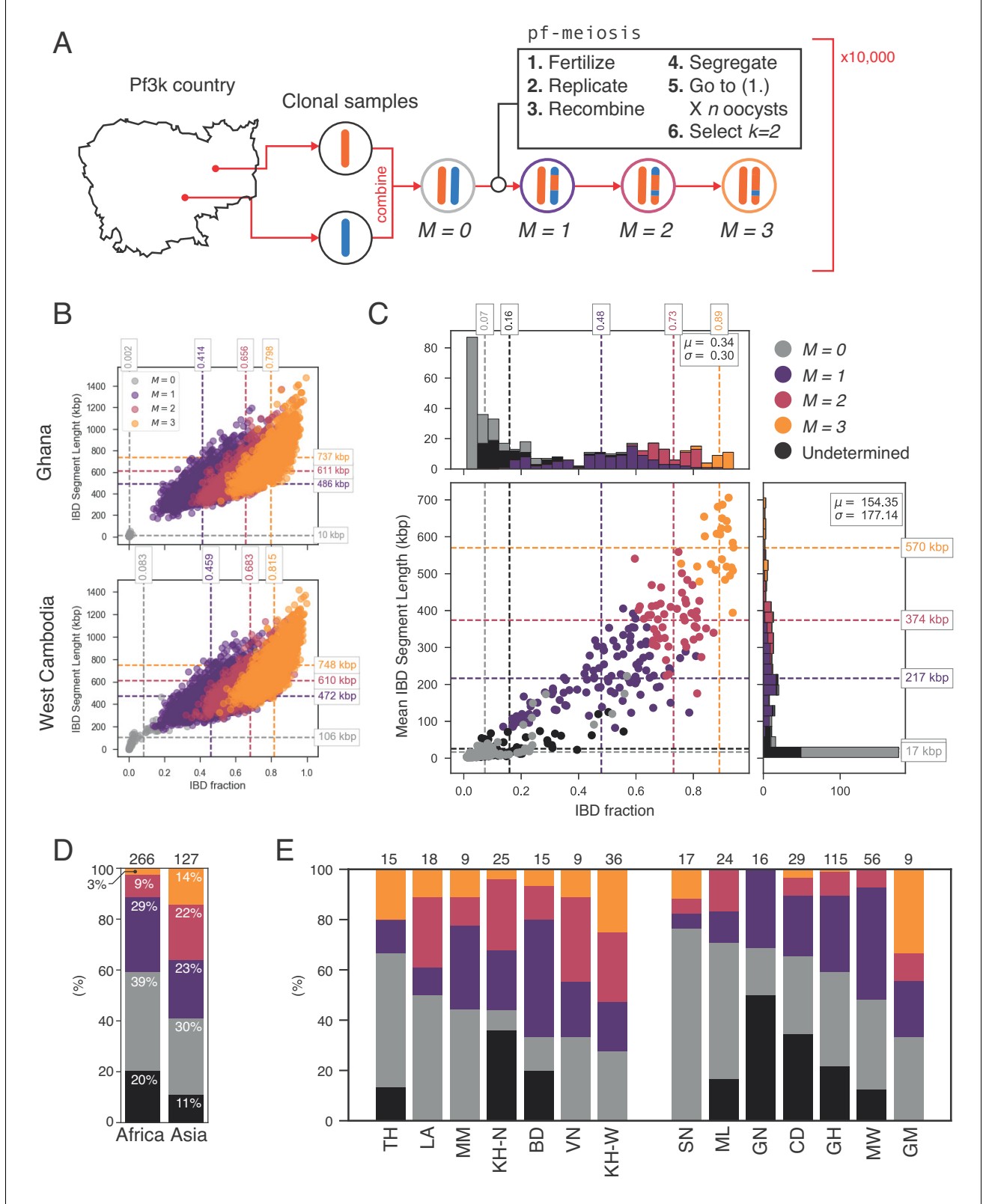

**Figure 6.** Identifying sibling strains within mixed infections. (**A**) Schematic showing how IBD fraction and IBD segment length distributions are created for $k = 2$ mixed infections using pf-meiosis. Two clonal samples from a given country are combined to create an unrelated ($M = 0$, where $M$ is number of meioses that have occurred) mixed infection. The $M = 0$ infection is then passed through 3 rounds of pf-meioses to generate $M = 1, 2, 3$ classes, representing serial transmission of the mixed infection ($M = 1$ are siblings). (**B**) Simulated IBD distributions for $M = 0, 1, 2, 3$ for Ghana (top) and

*Figure 6 continued on next page*

*Figure 6 continued*

West Cambodia (bottom). A total of 10,000 mixed infections are simulated for each class, from 500 random pairs of clonal samples. (**C**) Classification results for 393 $K = 2$ mixed infections from 13 countries. Undetermined indicates mixed infections with IBD statistics that were never observed in simulation. (**D**) Breakdown of class percentage by continent. Total number of samples is given above bars. Colours as in panel C ($M = 0$, grey; $M = 1$, purple; $M = 2$, pink; $M = 3$, orange; Undetermined, black). (**E**) Same as (**D**), but by country. Abbreviations as in *Figure 4*.

DOI: https://doi.org/10.7554/eLife.40845.016

a formidable challenge. Previously we developed an algorithm, `DEploid`, for deconvoluting mixed infections (*Zhu et al., 2018d*). However, we subsequently noticed the presence of mixed infections with highly related strains in which the algorithm performed poorly, particularly with low-frequency minor strains. Mixed infections containing highly related strains represent an epidemiological scenario of particular interest, because they are likely to have been produced from a single mosquito bite, itself multiply infected, and in which meiosis has occurred to generate sibling strains. Thus, we developed an enhanced method, `DEploidIBD`, capable not only of deconvoluting highly related mixed infections, but also inferring IBD segments between all pairs of strains present in the infection. Validation work using simulated mixed infections illustrated that `DEploidIBD` performs well on infections of two or three strains and across a wide-range of IBD levels. We note that limitations and technical difficulties remain, including deconvoluting infections with more than three strains, handling mixed infections with highly symmetrical or asymmetrical strain proportions (e.g. $K = 3$ with

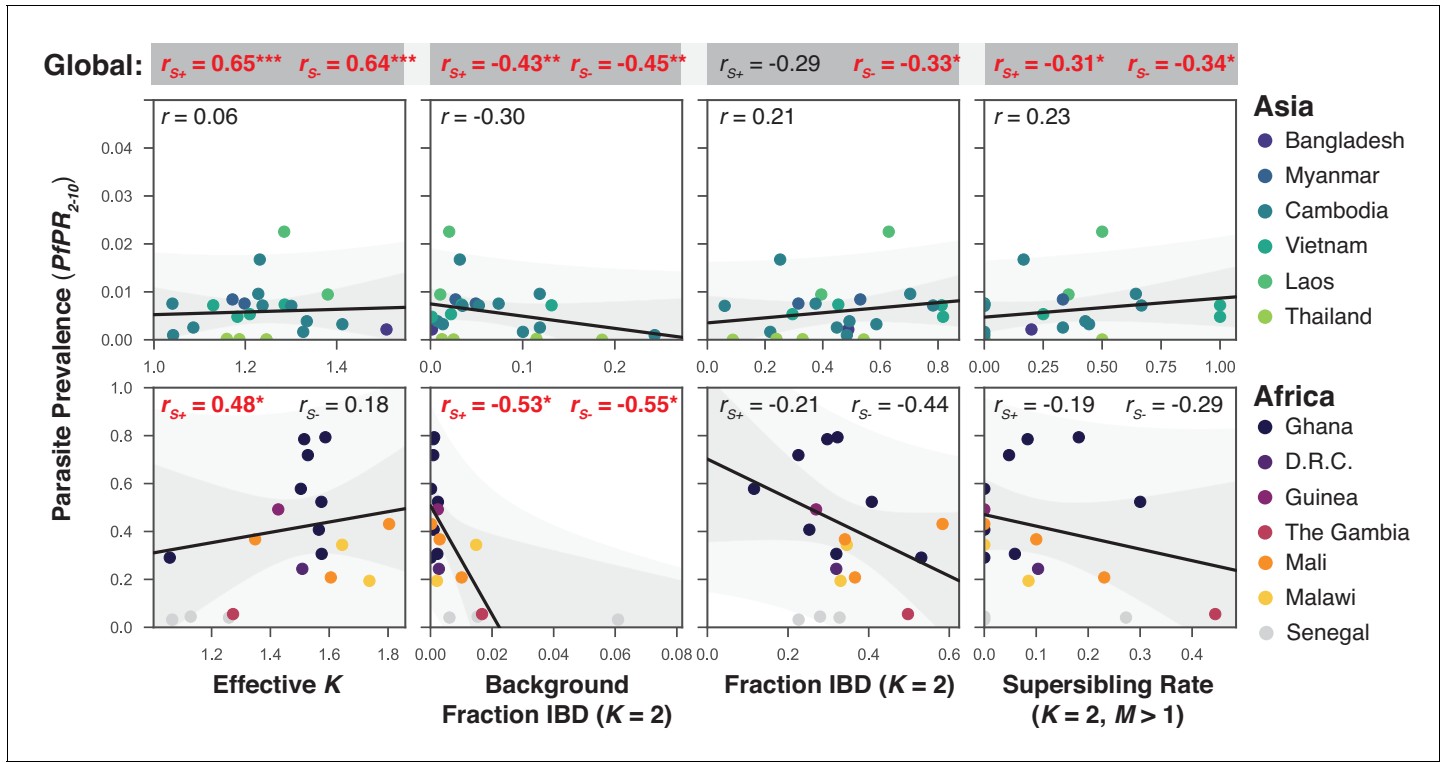

**Figure 7.** The relationship between *P. falciparum* prevalence and characteristics of mixed infection. Four mixed infection statistics are shown including the average effective number of strains (Effective K, first column), given by $K_e = (\sum w_i^2)^{-1}$, where $w_i$ is the proportion of the $i$th strain; background IBD observed between clonal samples (Background Fraction IBD, second column); fraction IBD within $K = 2$ mixed infections (Fraction IBD, third column); and the rate of $K = 2$ mixed infections classified as having $M > 1$ (Supersibling Rate, fourth column). Each point relates to a row in *Table 1* from different sampling locations and years. Pearson's r is computed globally (shown at top in a grey box for each statistic), across Asian countries (upper panel) and across African countries (lower panel). Globally and for Africa, the correlations were computed including Senegal ($r_{S+}$) and excluding Senegal ($r_{S-}$). The slope and confidence intervals for the regression line excluding Senegal are drawn. Significant correlations ($p < 0.05$) are highlighted in red and significance levels indicated by asterisks (* <0.05, ** <0.01, *** <0.001).

DOI: https://doi.org/10.7554/eLife.40845.017

strains at 33%, or $K = 2$ with one strain at 2%), analysing data with multiple infecting species, coping with low-coverage data, and selecting appropriate reference panels from the growing reference resources.

The application of `DEploidIBD` to the 2344 samples in the Pf3k project has revealed the extent and structure of relatedness among malaria infections and how these characteristics vary between geographic locations. We found that 1026 (44%) of all samples in Pf3k were mixed, being comprised of 480 $K = 2$ infections, 372 $K = 3$ and 127 $K = 4$ infections. Across the entire data set, the total number of genomes extracted from mixed infections is nearly double the number extracted from clonal infections (2584 genomes from $K>1$ vs. 1365 from $K = 1$). We also found considerable variation, between countries and continents in the characteristics of mixed infections, suggesting that they are sensitive to local epidemiology. Previous work has highlighted the utility of mixed infection rate in discerning changes in regional prevalence, and we re-enforce that finding here, observing a significant correlation between the effective number of strains and parasite prevalence across Pf3k collection sites. Similarly, using `DEploidIBD` we also observe significant geographical variation in the relatedness profiles of strains within mixed infections. Interestingly, this variation is structured such that regions with high rates of mixed infection tend to contain strains that are less related, resulting in a significant negative correlation between mixed infection rate and mean relatedness within those infections.

The ability to identify the extent and genomic structure of IBD enables inference of the mechanisms by which mixed infections can arise. A mixed infection of $K$ strains can be produced by either $K$ independent infectious bites or by $j<K$ infectious bites. In the first case, parasites are delivered by separate vectors and no meiosis occurs between the distinct strains, thus any IBD observed in the mixed infection must have pre-existed as background IBD between the individual strains. In the second case, meiosis may occur between strains, resulting in long tracts of IBD. The exact amount of IBD produced by meiosis is a random variable, dependent on outcomes of meiotic processes, such as the number of recombination events, the distance between them, and the segregation of chromosomes. Importantly, the mean IBD produced during meiosis in *P. falciparum* also depends on the number and type (selfed vs outbred) of oocysts in the infectious mosquito. Here, we have shown, from first principles, that the amount of IBD expected in a single-bite mixed infection produced from two unrelated parasites strains will always be slightly less than 1/2, and potentially as low as 1/3 (see Appendix).

To quantify the distribution of IBD statistics expected through different mechanisms of mixed infection, we developed a Monte Carlo simulation tool, `pf-meiosis`, which we used to infer the recent transmission history of individuals with dual ($K = 2$) infections. We considered mixed infection chains, in which $M$ successive rounds of meiosis, transmission to host, and uptake by vector can result in sibling strain infections with very high levels of IBD. Using this approach, we found that 47% of dual infections within the Pf3k Project likely arose through co-transmission events. Moreover, and particularly within Asian population samples, we found evidence for long mixed infection chains ($M>1$), representing repeated co-transmission without intervening superinfection. This observation is not a product of lower genetic diversity in Asia, as differences in background IBD between countries have been controlled for in the simulations. Rather, it reflects true differences in transmission epidemiology between continents. These findings have three important consequences. Firstly, they suggest that successful establishment of multiple strains through a single infection event is a major source of mixed infection. Second, they imply that the bottlenecks imposed at transmission (to host and vector) are relatively weak. Finally, they indicate that the differing mechanisms causing mixed infections reflect aspects of local epidemiology.

We note that a non-trivial fraction (17%) of all mixed infections had patterns of IBD inconsistent with the simulations (typically with slightly higher IBD levels than background but lower than among siblings). We suggest three possible explanations. A first is that the unclassified samples result from the IBD profiles produced by `DEploidIBD`, in particular the overestimation of short IBD tracks, similar to the issue observed by *Wong et al. (2018)*. Alternatively, our estimate of background IBD, generated by combining pairs of random clonal samples from a given country into an artificial $M = 0$ mixed infection, will underestimate true background IBD if there is very strong local population structure. Finally, we only simulated simple mixed infection transmission chains, at the exclusion of more complex transmission histories, such as those involving strains related at the level of cousins. The extent to which such complex histories can be inferred with certainty remains to be explored.

Lastly, our results show that the rate and relatedness structure of mixed infections correlate with estimated levels of parasite prevalence, at least within Africa, where prevalence is typically high (*Smith et al., 1993*). In Asia, which has much lower overall prevalence, as well as greater temporal (and possibly spatial) fluctuations, we do not observe such correlations. However, it may well be that other genomic features that we do not consider in this work could provide much higher resolution, in space and time, for capturing changes in prevalence than traditional methods. Testing this hypothesis will lead to a much greater understanding of how genomic data can potentially be used to inform global efforts to control and eradicate malaria.

## Materials and methods

The data analysed within this paper were collected and made openly available to researchers by member of the Pf3k Consortium. Information about studies within the data set can be found at https://www.malariagen.net/projects/pf3k#sampling-locations. Detailed information about data processing can be found at https://www.malariagen.net/data/pf3k-5. Briefly, field isolates were sequenced to an average read depth of 86 (range 12.6–192.5). After removing human-derived reads and mapping to the 3D7 reference genome, variants were called using GATK best practice and approximately one million variant sites were genotyped in each isolate. After filtering samples for low coverage and cross-species contamination, 2344 samples remained. The Appendix provides details on the filters used and data availability. For deconvolution, samples were grouped into geographical regions by genetic similarity; four in Africa, and three in Asia. (*Table 1*). Reference panels were constructed from the clonal samples found at each region. Since previous research has uncovered severe population structure in Cambodia (*Miotto et al., 2013*), we stratified samples into West and North Cambodia when performing analysis at the country level.

### Data availability

Metadata on samples is available from ftp://ngs.sanger.ac.uk/production/pf3k/release_5/pf3k_release_5_metadata_20170804.txt.gz. Sequence data (aligned to *Plasmodium falciparum* strain 3D7 v3.1 reference genome sequences, for details see ftp://ftp.sanger.ac.uk/pub/project/pathogens/gff3/2015-08/Pfalciparum.genome.fasta.gz) is available from ftp://ngs.sanger.ac.uk/production/pf3k/release_5/5.1/. Diagnostic plots for the deconvolution of all samples can be found at https://github.com/mcveanlab/mixedIBD-Supplement (*Zhu, 2018a*; copy archived at https://github.com/elifesciences-publications/mixedIBD-Supplement) and deconvoluted haplotypes can be accessed at ftp://ngs.sanger.ac.uk/production/pf3k/technical_working/release_5/mixedIBD_paper_haplotypes/. Code implementing the algorithms described in this paper, `DEploidIBD`, is available at https://github.com/DEploid-dev/DEploid (*Zhu, 2018b*; copy archived at https://github.com/elifesciences-publications/DEploid). Code to generate in silico lab mixture of 4 strains are available at https://github.com/DEploid-dev/DEploid-Data-Benchmark-in_silico_lab_mixed_4s (*Zhu, 2019b*; copy archived at https://github.com/elifesciences-publications/DEploid-Data-Benchmark-in_silico_lab_mixed_4s).

Code to generate in silico field mixtures of 2, 3, four strains are available at https://github.com/DEploid-dev/DEploid-Data-Benchmark-in_silico_field (*Zhu, 2019a*; copy archived at https://github.com/elifesciences-publications/DEploid-Data-Benchmark-in_silico_field).

## Additional information

### Funding

| Funder | Grant reference number | Author |
| --- | --- | --- |
| Wellcome | 206194 | Jacob Almagro-Garcia |
| Wellcome | 090770 | Jacob Almagro-Garcia<br>Richard D Pearson<br>Roberto Amato<br>Alistair Miles<br>Dominic Kwiatkowski |
| Wellcome | 100956/Z/13/Z | Sha Joe Zhu<br>Gil McVean |

| | | |
|---|---|---|
| Li Ka Shing Foundation | | Gil McVean |
| Wellcome | 204911 | Jacob Almagro-Garcia<br>Richard D Pearson<br>Roberto Amato<br>Alistair Miles<br>Dominic Kwiatkowski |
| Medical Research Council | G0600718 | Jacob Almagro-Garcia<br>Richard D. Pearson<br>Roberto Amato<br>Alistair Miles<br>Dominic Kwiatkowski |
| Department for International Development | M006212 | Jacob Almagro-Garcia<br>Richard D Pearson<br>Roberto Amato<br>Alistair Miles<br>Dominic Kwiatkowski |

The funders had no role in study design, data collection and interpretation, or the decision to submit the work for publication.

## Author contributions

Sha Joe Zhu, Conceptualization, Software, Formal analysis, Validation, Investigation, Visualization, Methodology, Writing—original draft, Writing—review and editing; Jason A Hendry, Conceptualization, Software, Formal analysis, Validation, Investigation, Visualization, Methodology, Writing—review and editing; Jacob Almagro-Garcia, Software, Formal analysis, Validation, Investigation, Methodology, Writing—review and editing; Richard D Pearson, Roberto Amato, Daniel J Weiss, Tim CD Lucas, Michele Nguyen, Peter W Gething, Data curation, Writing—review and editing; Alistair Miles, Conceptualization, Writing—review and editing, Development of the simulation pf-meiosis, Vector biology; Dominic Kwiatkowski, Conceptualization, Supervision, Writing—review and editing; Gil McVean, Conceptualization, Resources, Data curation, Software, Formal analysis, Supervision, Investigation, Visualization, Methodology, Writing—original draft, Writing—review and editing

## Author ORCIDs

Sha Joe Zhu (iD) https://orcid.org/0000-0001-7566-2787
Jason A Hendry (iD) https://orcid.org/0000-0003-4164-3179
Jacob Almagro-Garcia (iD) https://orcid.org/0000-0002-0595-7333
Richard D Pearson (iD) https://orcid.org/0000-0002-7386-3566
Gil McVean (iD) https://orcid.org/0000-0002-5012-4162

## Decision letter and Author response

Decision letter https://doi.org/10.7554/eLife.40845.039
Author response https://doi.org/10.7554/eLife.40845.040

# Additional files

## Supplementary files

• Supplementary file 1. About the Pf3k Project.
DOI: https://doi.org/10.7554/eLife.40845.018

• Transparent reporting form
DOI: https://doi.org/10.7554/eLife.40845.019

## Data availability

Metadata on samples is available from ftp://ngs.sanger.ac.uk/production/pf3k/release_5/pf3k_release_5_metadata_20170804.txt.gz. Sequence data (aligned to Plasmodium falciparum strain 3D7 v3.1 reference genome sequences, for details see ftp://ftp.sanger.ac.uk/pub/project/pathogens/gff3/2015-08/Pfalciparum.genome.fasta.gz) is available from ftp://ngs.sanger.ac.uk/production/pf3k/release_5/5.1/. Diagnostic plots for the deconvolution of all samples can be found at https://github.

com/mcveanlab/mixedIBD-Supplement (copy archived at https://github.com/elifesciences-publications/mixedIBD-Supplement) and deconvolved haplotypes can be accessed at ftp://ngs.sanger.ac.uk/production/pf3k/technical_working/release_5/mixedIBD_paper_haplotypes/. Code implementing the algorithms described in this paper, DEploidIBD, is available at https://github.com/mcveanlab/DEploid (copy archived at https://github.com/elifesciences-publications/DEploid).

The following previously published dataset was used:

| Author(s) | Year | Dataset title | Dataset URL | Database and Identifier |
|---|---|---|---|---|
| The Pf3k Project Consortium | 2016 | The Pf3k Project (2016): pilot data release 5 | http://www.malariagen.net/data/pf3k-5 | Wellcome Trust Sanger public ftp site, 5.1 Data |

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

## Appendix 1

DOI: https://doi.org/10.7554/eLife.40845.020

# Deconvolution in the presence of IBD

## Notation

We use the same notations as *Zhu et al. (2018d)* (see *Appendix 1—table 1*). Our data, $D$, are the allele read counts of sample $j$ at a given site $i$, denoted as $r_{j,i}$ and $a_{j,i}$ for reference (REF) and alternative (ALT) alleles respectively. These have assigned values of 0 and 1, respectively. Here we consider only bi-allelic loci, though future extensions to the model to include multi-allelic sites could be accommodated. The empirical allele frequencies within a sample (WSAF) $p_{j,i}$ and at population level (PLAF) $f_i$ are calculated by $\frac{a_{j,i}}{a_{j,i}+r_{j,i}}$ and $\frac{\sum_j a_{j,i}}{\sum_j a_{j,i}+\sum_j r_{j,i}}$ respectively. Since all data in this section refers to the same sample, we drop the subscript $j$ from now on.

## Modelling mixed infections with IBD

We describe the mixed infection problem by considering the number of strains, $K$, the relative abundance of each strain, $\mathbf{w}$, and the allelic state of the $k$th strain at the $i$th locus, $\mathbf{h}_{k,i}$. In addition to *Zhu et al. (2018d)*, we also infer the identity-by-descent (IBD) state $\mathcal{S}_i$, which describes the strain relationship with each other at site $i$. For example, for three strains, the IBD-state could be one of the five cases: (1) all strains are not IBD; (2) strains 1 and 2 are IBD; (3) strains 1 and 3 are IBD; (4) strains 2 and 3 are IBD; (5) all strains are IBD (see *Appendix 1—table 2*). To simplify the problem, we assume independence between markers, and drop the subscript $i$ from now on. As previously, we use a Bayesian approach and define the posterior probabilities of $K$, $\mathbf{w}$, $\mathbf{h}$ and $\mathcal{S}$, as

$$P(K, \mathbf{w}, \mathbf{h}, \mathcal{S}|e, D) \propto L(K, \mathbf{w}, \mathbf{h}, \mathcal{S}|e, D) \times P(K, \mathbf{w}, \mathbf{h}, \mathcal{S}), \tag{1}$$

where $e$ is the read error rate. Moreover, we can decompose the joint prior as

$$P(K, \mathbf{w}, \mathbf{h}, \mathcal{S}) = P(K) \times P(\mathbf{w}|K) \times P(\mathcal{S}|K) \times P(\mathbf{h}|\mathcal{S}). \tag{2}$$

The number of strains, $K$, and their proportions $\mathbf{w}$, are as in *Zhu et al. (2018d)*: $K$ is fixed at a user-defined value (here, $K = 4$) and strains below a fixed proportion threshold (here, 0.01) are removed. To model IBD configurations and resulting haplotypes we introduce a parameter $\theta$, which is the probability of outbreeding (i.e. non-IBD) for a mixed infection with two strains. Specifically, for $K$ strains, the prior probability on configuration $\mathcal{S}$ is a function of the number of distinct strains in the configuration, $C_{\mathcal{S}}$, and the number of distinct configurations with the same number of distinct strains, $M_{\mathcal{C}}$:

$$P(\mathcal{S}|K) = \binom{K-1}{C_{\mathcal{S}}-1} \frac{\theta^{C_{\mathcal{S}}-1}(1-\theta)^{K-C_{\mathcal{S}}}}{M_{\mathcal{C}}}. \tag{3}$$

That is, the number of distinct strains is drawn from a binomial distribution with parameters $K$ and $\theta$ and the IBD configuration is uniformly drawn from among those with the same number of distinct strains. Consequently, the expected number of distinct haplotypes (i.e. non-IBD) in an infection with $K$ strains is $1 + \theta(K - 1)$. The value of $\theta$ estimated within the Markov chain Monte Carlo (see details below).

Suppose that the probability of recombination event between two adjacent sites is $p_{rec}$ (see Implementation details). To model the transitions between IBD states we make the approximation that the IBD state at the $i + 1$th site is the same as at the $i$th site, with probability $1 - p_{rec}$ and, if there is a recombination event, the IBD state is drawn from the stationary distribution described above.

To model allelic states (or genotype), conditional on IBD state, we assume that allelic states for each IBD group are independent Bernoulli draws given the population allele frequency (PLAF). For example, if $K = 3$ and only strains 1 and 2 are IBD (state 0,0,1), the genotype at site $i$, $\mathbf{h_i}$, could be $\{0, 0, 0\}$ or $\{1, 1, 1\}$, with probabilities $(1 - f)^2$, $(1 - f)f$, $f(1 - f)$ and $f^2$ respectively. The joint prior on IBD and allelic states is referred to as $\psi(\mathcal{S}, \mathbf{h})$. Note that we ignore association (known as linkage disequilibrium or LD) between nearby alleles, though note that in implementation we run `DEploidIBD` twice; first as described here to estimate strain number and proportions, allowing for IBD, then subsequently with a reference panel, as for `DEploid`, but with the strain number and proportions fixed. This latter step does model LD and consequently results in better estimates of strain haplotypes.

Details of the emission model are the same as described in *Zhu et al. (2018d)*. Briefly, the reference and alternative allele read counts at each site are modelled as being drawn from a beta-binomial distribution given the expected WSAF, $q = \mathbf{w}^T\mathbf{h}$. To incorporate sequencing error, we modify the expected WSAF such that allele are misread with probability $e$. Thus, the adjusted expected WSAF becomes

$$\pi_i = q_i + (1 - q_i)e - q_ie = q_i + (1 - 2q_i)e. \tag{4}$$

As previously, we model over-dispersion in read counts relative to the Binomial using a Beta-binomial distribution.

## Parameter estimation using Markov chain Monte Carlo

To infer the relatedness between strains within a mixed infection and their relative proportions we use a Markov chain Monte Carlo (MCMC) approach. We learn the relative abundance of each strain by exploiting signatures of within-sample allele frequency imbalance, using a Metropolis-Hastings algorithm, which samples proportions, $\mathbf{w}$, given IBD-configurations, $\mathcal{S}$. We then use a Gibbs sampler to update $\mathcal{S}$ and $\mathbf{h}$ for a given $\mathbf{w}$ by first sampling the IBD state from the posterior, and then sampling the allelic configuration (genotype) from the selected IBD state at each site to update haplotypes. Note that the hidden Markov model structure for IBD states along the chromosome leads to efficient algorithms for calculating quantities of importance. Notably, by summing over allelic configurations that are consistent with a given IBD configuration, we can - for a given set of strain proportions - calculate the likelihood integrated over all possible IBD configurations, which leads to efficient sampling.

### Metropolis-Hastings update for proportions

We update strain proportions, $\mathbf{w}$, through modelling underlying log titres, $\mathbf{x}$, where $w_i \propto exp(x_i)$. As previously, log titres are assumed to be drawn from a $N(-log(K), \sigma^2)$ distribution under the prior. To update, we choose $i$ uniformly from $K$ and propose new $x_i'$s from $x_i' = x_i + \delta x$, where $\delta x \sim N(0, \sigma^2/s)$, and $s$ is a scaling factor. The new proposed proportion is therefore $\frac{exp(x_i')}{\sum_{k=1}^{K} exp(x_k')}$. Since the proposal distribution is symmetrical, the Hastings ratio is 1. The proposed update is accepted with probability

$$\min\left(1, \frac{P(\mathbf{w}'|K)}{P(\mathbf{w}|K)}\frac{L(\mathbf{w}', \mathbf{h}|e, D)}{L(\mathbf{w}, \mathbf{h}, |e, D)}\right).$$

Note that, conditional on the current estimate of strain haplotypes, $\mathbf{h}$, the likelihood is not a function of the specific IBD configuration.

### Gibbs sampling update for haplotype and IBD-configuration update

As described above, in `DEploidIBD`, allelic states at different sites within a haplotype are independent. However, IBD states are connected through a Markov process. We therefore update the IBD configuration and strain haplotypes in a two stage process. First, we use the Forward algorithm to first compute the integrated likelihood - that is the probability of observing the read-level data integrating over all possible IBD configurations and allelic states.

Defining $F_{i,\mathcal{S},\mathbf{h}}$ to be the integrated likelihood for the set of paths ending in IBD configuration $\mathcal{S}$ at site $i$ and with allelic configuration $\mathbf{h}$, it follows that

$$F_{i,\mathcal{S},\mathbf{h}} = P(D_i|e,\mathbf{h})[(1-p_{rec})F_{i-1,\mathcal{S},\mathbf{h}} + \psi(\mathcal{S},\mathbf{h})p_{rec}\sum_{m,n}F_{i-1,m,n}], \qquad (5)$$

where $\psi(\mathcal{S},\mathbf{h})$ is the prior on the combination of IBD configuration $\mathcal{S}$ and allelic configuration $\mathbf{h}$ as defined in Section (Modelling mixed infections with IBD). Note that the summation term is identical for all IBD/allelic configurations.

By storing the output of the Forward algorithm we can then sample from the posterior distribution of the IBD- and allelic-configurations (given current proportions). That is, we sample $\mathcal{S}_l, \mathbf{h}_l \propto F_{l,\mathcal{S},\mathbf{h}}$, and then previous steps proportion to $F_{i,\mathcal{S},\mathbf{h}}$ times the probability to transition to the sampled state at position $i+1$.

## Caveat: identifiability with balanced mixing

Using a reference panel free approach means that it can be difficult or impossible to deconvolute samples containing strains with equal proportions. For example, it is impossible to distinguish between $\{\frac{1}{3},\frac{1}{3},\frac{1}{3}\}$ and $\{\frac{1}{3},\frac{2}{3}\}$, or $\{\frac{1}{4},\frac{1}{4},\frac{1}{4},\frac{1}{4}\}$ and $\{\frac{1}{4},\frac{1}{4},\frac{1}{2}\}$. Because of the prior we use, the model will prefer to merge haplotypes when possible, which can lead to underestimation of the number of strains. We advise users to apply DEploid with multiple runs with and without the '-ibd' flag and see if such problem occurs.

## Gibbs sampling update for IBD parameter

Given the sampled IBD configuration, the IBD parameter $\theta$ can be updated using Gibbs sampling. We first identify the path of the $D$ distinct IBD configurations along the chromosome and, for each, identify the number of distinct IBD groups, $\mathcal{C}_d$. For example, if the IBD state is 0,1,2,2 there are three distinct IBD groups (0,1,2; $\mathcal{C}_d = 3$), while if the IBD state is 0,0,1,1 there are two IBD groups (0,1; $\mathcal{C}_d = 2$). Under the prior, the number of distinct strains (minus 1) is drawn from a binomial distribution with parameters $K-1$ and $\theta$. With a uniform prior on $\theta$, a new value, $\theta'$ is therefore drawn from:

$$\theta' \sim Beta(\mathcal{C}_D - D + 1, DK - \mathcal{C}_D + 1),$$

where $\mathcal{C}_D = \sum_{d=1}^{D}\mathcal{C}_d$.

## Implementation details

Below we detail a number of implementation details.

### Approximating the likelihood surface

At any site, the likelihood for the WSAF, $q$, induced by the allelic configuration and strain proportions derives from the beta-binomial model as implemented in DEploid. That is, the likelihood of $q_i$ is

$$L(q_i|e,D) \propto \frac{B(a_i + \pi c, r_i + (1-\pi)c)}{B(\pi c, (1-\pi)c)}, \qquad (6)$$

where $\pi_i$ is the adjusted WSAF $\pi = q(1-e) + (1-q)e$ and $c^{-1}$ determines the magnitude of over-dispersion relative to the binomial, with $c \to \infty$ recovering the binomial. Here, we use $c = 100$.

For computational efficiency, rather than recomputing for every value of $q$, we first approximate the likelihood function for each site through a Beta distribution, matching the first two moments of the posterior on $q$ implied by **Equation 6** within a uniform prior on $q$. The estimated parameters of the Beta distribution are then used to approximate the likelihood surface in all subsequent calculations.

## MCMC parameters for deconvolution

- Number of strains. As described above, we aim to infer more strains than are actually present, starting the MCMC chain with a fixed $K$ (default of 5). In our experience, we find poor performance for $K>4$, hence use the flag -k four to specify the number of strains as 4. At the point of reporting, we keep strains with a proportion above a fixed threshold, typically 0.01.

- Parameters. We typically set the log-titre variance $\sigma^2 = 5$, which can be adjusted when working with extremely unbalanced samples (see **Zhu et al., 2018d**, Supplementary Material). We set the read error rate as 0.01 and the rate of mis-copying as 0.01.

- Recombination rate and scaling. We assume a uniform recombination map, where the genetic distance between loci $i$ and $i+1$ is computed by $\gamma_i = D_i/d_m$ where $D_i$ denotes the physical distance between loci $i$ and $i+1$ in nucleotides and $d_m$ denotes the average recombination rate in Morgans $bp^{-1}$. We use the recombination rate for *P. falciparum* of 13,500 base pairs per centiMorgan as reported by **Miles et al. (2016)**. The recombination rate is scaled by a factor $G$, which reflects the effective population size, rate of inbreeding, and relatedness of the reference panel. In practice, we deconvolute over 1 million markers in field samples. We tuned the parameter $G$ using in vitro lab mixtures, finding that a value of $G = 20$ ensures small recombination probabilities between any two markers, with a mean of 0.015. A large value of $G$ relaxes the reference panel constraint, becoming an LD free model when $G$ is infinity. The scaled genetic distance $G\gamma$ is used to compute the transition probability of switching from copying reference haplotype $a$ to reference haplotype $b$. Given LD varies enormously between *P. falciparum* populations, we will investigate how to tune this parameter for future improvement. In the current release, our program allows users to apply the flag -G to specify a specific value. We note that IBD deconvolution from error-prone data can benefit from even smaller values of $G$, such as 0.1.

- Reporting. We aim to provide users with a single point estimate of the haplotypes, their proportions and IBD configurations, although the full chain is also available for analysis. To achieve this we report values at the last iteration - that is we report a single sample from the posterior. However, to measure robustness, we typically repeat the deconvolution with multiple random starting points. We use a majority vote rule on the inferred number of strains; we then select the chain with the lowest average deviance (after removing the burn-in) as our estimate. The deviance measures the difference in log likelihood between the fitted and saturated models, the latter being inferred by setting the WSAF to that of the observed values. These parameters can be modified by users to achieve a preferred balance between computational speed and confidence. By default, we set the MCMC sampling rate as 5, with the first 50% of samples removed as burn in and 800 samples used for estimation.

- Reference panel construction. To infer clonal samples for the reference panel we use the Pf3k project data, running the algorithm without LD on all samples and identifying those with a dominant haplotype (proportion > 0.99) as clonal. These clonal samples are grouped by region of sampling to form location-specific reference panels. In addition, we have included a number of reference strains, described in more detail below.

## Summarising pairwise IBD

At the end of a run, we obtain the maximum likelihood estimate of IBD configurations along the chromosome using the Viterbi algorithm. Patterns of IBD are then obtained for all pairs of strains and summarised through the mean fraction IBD and the N50 IBD tract, obtained by identifying all blocks of IBD, sorting them in decreasing size and finding the size of the block such that 50% of all sites that are in IBD lie in blocks of at least this size. A larger N50 statistic indicates more recent common ancestry.

We note that it is also possible to obtain estimates of pairwise IBD from the full posterior distribution of pairwise IBD (using standard hidden Markov model practices). Typically these give very similar answers to the Viterbi solution, though are typically slightly larger due to the identification of low certainty regions. The posterior mean IBD between pairs is also provided to users as output.

## Commands

Each isolate deconvolution is repeated 15 times, each time initialized with a random seed. For each run we obtain an estimate of the number of strains and we take the modal value to be the estimate. For reporting, we use the first run that estimated the consensus number of strains. The text below gives an example of an input script for deconvolution. Full details are available in the documentation at the Github page: https://github.com/mcveanlab/DEploid/ (**Zhu, 2018b**; copy archived at https://github.com/elifesciences-publications/DEploid).

```
ref = PD0577 C_ref.txt
alt = PD0577 C_alt.txt
plaf = asiaGroup1_PLAF.txt
panel = asiaGroup1PanelMostDiverse10.csv
exludeAt = asiaGroup1_and_pf3k_bad_snp_in_at_least_50_samples.txt
prefix = PD0577 C_IBD time dEploid -ref ${ref} -alt ${alt} -plaf ${plaf} -panel
${panel} \ -exclude ${exludeAt} -o ${prefix} -nSample 250 -rate 8 -burn 0.67 -
ibd -k 4 interpretDEploid.r -ref ${ref} -alt ${alt} -plaf ${plaf} -o ${prefix} \
-dEprefix ${prefix} -exclude ${exludeAt}
```

## Error analysis

For haplotype quality analysis, we compared the inferred haplotype (`DEploid` output) with the true haplotype. In addition to mismatches between the testing haplotype and the truth, the deconvolution process also introduces switch errors, where the haplotypes are correct, but have undergone in silico recombination events. In addition, when one strain has low frequency, there might be insufficient data to make accurate inference, resulting in missing segments of the true haplotypes. We refer to this as dropout error. Dropout error can also be caused by the sequencing process, when parts of the genome are not well sequenced (low read count).

When comparing inferred haplotypes to true haplotypes we use a dynamic programming approach to find a description of the differences, optimising a cost function in which switch errors are twice as costly as mismatches and allele dropouts. Code for performing the optimal description, `errorAnalysis.r`, is available at the Github page: https://github.com/DEploid-dev/DEploid-Utilities/(**Zhu, 2018c**; copy archived at https://github.com/elifesciences-publications/DEploid-Utilities). An example analysis for three strains is shown in **Appendix 1—figure 1**.

**Appendix 1—table 1.** Notation used in this article.

| | |
|---|---|
| $i$ | Marker index |
| $j$ | Sample index |
| $r$ | Read count for reference allele |
| $a$ | Read count for alternative allele |
| $f$ | Population level allele frequency (PLAF) |
| $K$ | Number of distinct strains within sample |
| $l$ | Number of sites |
| $\mathbf{w}$ | Proportions of strains |
| $\mathbf{x}$ | Log titre of strains |
| $\mathbf{h}_i$ | Allelic states of $K$ parasite strains at site $i$ |
| $h_{k,i}$ | Allelic state of parasite strain $k$ at site $i$ |
| $p$ | Observed within sample allele frequency (WSAF) |
| $q$ | Unadjusted expected WSAF |
| $\pi$ | Adjusted expected WSAF |

*Appendix 1—table 1 continued on next page*

| | |
|---|---|
| $e$ | Probability of read error |
| $S_i$ | IBD configuration at site $i$ |
| $\theta$ | Probability of non-IBD in a mixture of two strains |

DOI: https://doi.org/10.7554/eLife.40845.021

**Appendix 1—table 2.** IBD configurations for two, three and four strains, ordered top to bottom by the number of IBD pairs. The (zero-indexed) notation indicates the type assigned to each haplotype, thus 0–1 indicates non-IBD for two strains, while 0-1-2-2 indicates four strains in which the third and fourth are IBD.

| Index | IBD state | | |
|---|---|---|---|
| | K = 2 | K = 3 | K = 4 |
| 0 | 0–1 | 0-1-2 | 0-1-2-3 |
| 1 | 0–0 | 0-0-1 | 0-0-1-2 |
| 2 | | 0-1-0 | 0-1-0-2 |
| 3 | | 0-1-1 | 0-1-2-0 |
| 4 | | 0-0-0 | 0-1-1-2 |
| 5 | | | 0-1-2-1 |
| 6 | | | 0-1-2-2 |
| 7 | | | 0-0-1-1 |
| 8 | | | 0-1-0-1 |
| 9 | | | 0-1-1-0 |
| 10 | | | 0-0-0-1 |
| 11 | | | 0-0-1-0 |
| 12 | | | 0-1-0-0 |
| 13 | | | 0-1-1-1 |
| 14 | | | 0-0-0-0 |

DOI: https://doi.org/10.7554/eLife.40845.022

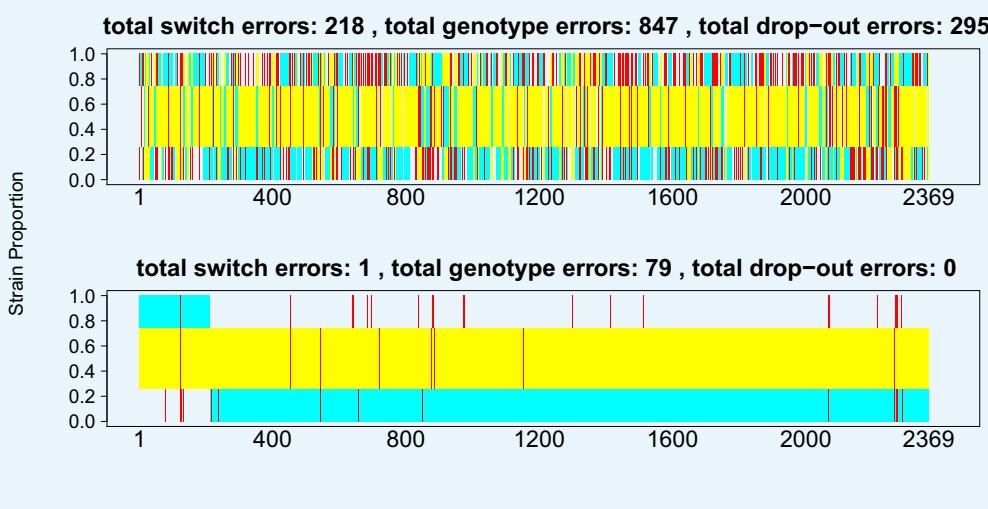

**Appendix 1—figure 1.** Comparison of true and inferred haplotypes for Chromosome 14 (2,369 SNPs) in the lab strain mixture sample PG0396-C after running `DEploidIBD` to infer strain number and proportions (top) and after subsequent refinement of haplotypes by running `DEploid` with Reference Panel V (bottom). The yellow, cyan and white backgrounds identify

the haplotype segments from strains 7G8, HB3 and Dd2 respectively. Numbers in the titles indicate the inferred switch, mismatch and dropout errors identified by the dynamic programming approach, with the cost of switch errors being twice that of other errors.
DOI: https://doi.org/10.7554/eLife.40845.023

# Appendix 2

DOI: https://doi.org/10.7554/eLife.40845.020

## Generating in silico mixtures for deconvolution benchmark

### In silico mixtures of lab strains

For in silico lab mixtures, we used the lab strain GB4 to provide a whole-genome read depth profile. We drew Bernoulli variables to simulate alternative allele counts, with the probability of success as the adjusted within sample allele frequency at each position (**Equation 4**). The expected WSAF is the dot product of the genotype of the lab strains: 3D7, Dd2, HB3, 7G8 and the proportions. For example, if the genotype is 0, 0, 1, one at an arbitrary position, and the mixture proportion of 0.1, 0.2, 0.4 and 0.3 will lead to an expected WSAF of 0.7. We then further adjust the WSAF with an error term (0.01) to account for all technical error/noise: $0.7 \times (1 - 0.01) + (1 - 0.7) \times .01 = 0.696$.

To test the accuracy of `DEploidIBD` in a more realistic setting, we created in silico mixtures of 2, 3 and 4 strains given different transmission scenarios from mosquito bites. Let $K$ denote the number strains within each sample, and $b$ denote the number of independent mosquito bites. In such a scenario, a mixed infection containing $K$ strains can be treated as a result of $b$-bite events, where $K \in \{2, 3, 4\}$ and $1 \sim \leq \sim b \sim \leq \sim K$. For example, a $K = 2$ mixed infection can result from co-transmission, where two strains are passed in a single bite ($b = 1$); or by superinfection, where each strain is delivered by a unique bite ($b = 2$). A mixed infection containing three strains is more complex: besides co-infection (three strains in a single bite, $b = 1$) and superinfection (3 strains in three bites, $b = 3$), we can also have a super-infection scenario made up from a co-infection plus a clonal-infection ($b = 2$). The complete simulation procedure (code) is available at https://github.com/DEploid-dev/DEploid-Data-Benchmark-in_silico_field.

### In silico mixtures of two field strains

To simulate a mixture of two strains, we randomly selected two strains from 189 clonal samples of African origin (proportions ranging from 10/90% to 50/50%) using Chromosome 14 data. A further 20 randomly chosen samples were used as the reference panel. In order to compare the accuracy of the two methods at different levels of relatedness, we set 0%, 25%, 50% and 75% of the second haplotype the same as the first haplotype to mimic scenarios of unrelated, low, medium and high relatedness respectively. This operation sets a lower limit to the relatedness between two strains, as background relatedness may also exist. We used empirical read depths and drew read counts for the two alleles from binomial proportions (the same approach for generating in silico lab mixtures). We excluded sites for analysis at zero alternative allele counts in both targeted samples and reference panel, kept and analysed around 7757 polymorphic sites (standard deviation 178) for each in silico samples.

We repeated the in silico experiment with mixtures of two strains from 204 clonal Asian samples, also with mixing proportions of ranging from 10/90% to 50/50%, using about 3041 sites (s.d. 227) from Chromosome 14.

### In silico mixtures of three field strains

We further extended benchmarking to in silico mixtures of three strains in African populations with mixing proportions of 10/10/80%, 10/25/65%, 15/25/60%, 10/40/50%, 15/30/55%, 20/30/50%, 33/33/34%. Two generate $b = 1$ infections with three strains, we randomly selected two strains, namely parent A and parent B, from 189 clonal samples of Africa origin, and set the first 33% of the first haplotype and the last 66% of the second haplotype the same as parent

A; the rest of the first and second haplotypes and the third haplotype the same as parent B, to mimic the scenario of a 1/3 pairwise relatedness within sample.

In addition to the basic co-infection and super-infection scenarios of 3 strains, we also consider more complex events when co-infection presents within a super-infection. This data simulation is similar to simulating 2 strains of 50% relatedness, with one additional unrelated haplotype. Therefore, the overall pairwise relatedness is $1/2$ distributed over three possible pairs, which leads to $1/6$.

## In silico mixtures of four field strains

We tested In silico deconvolution experiments on mixtures of 4 strains. From previous experiments, we have learned that strain compositions with even proportions are difficult to deconvolute. In this set of simulations, we experimented with various unbalanced and balanced proportions including 11/22/30/37%, 25/25/25/25%, 20/20/20/40% and 30/30/30/10%. For some cases, the data generation procedures can easily be modified from previous experiment: a $b = 4$ event of for four strains is equivalent to a 3-bite event of 3 strains with one extra random haplotype; a $b = 3$ event of four strains is equivalent to a 2-bite event of 3 strains with one extra random haplotype. For 2-bite event of 4 strains, there are two possibilities: (i) both bites pass on two strains, which is essentially repeating $b = 1$ event of 2 strains twice or (ii) three strains in one bite and one in another, which is essentially a 1-bite event of 3 strains with one extra random haplotype.

## Haplotype quality for in silico mixtures

We assessed the quality of all the haplotypes deconvolved with `DEploid` and `DEploidIBD` for the in silico simulated mixtures (**Appendix 2—figure 1**, **Appendix 2—figure 2**). Our results are consistent with the performance we observed in field samples. Complex mixtures with balanced proportions or marginal strains (i.e. with a very low proportion) tend to produce chimeric haplotypes. Nonetheless, further research is needed to explore factors that result in deconvolution failure. DEpoidIBD performs slightly worse than the previous version of the software; we ascribe this to the fact that the simulations lack any complex IBD structure. See (Haplotype quality assessment) for more details about the procedure.

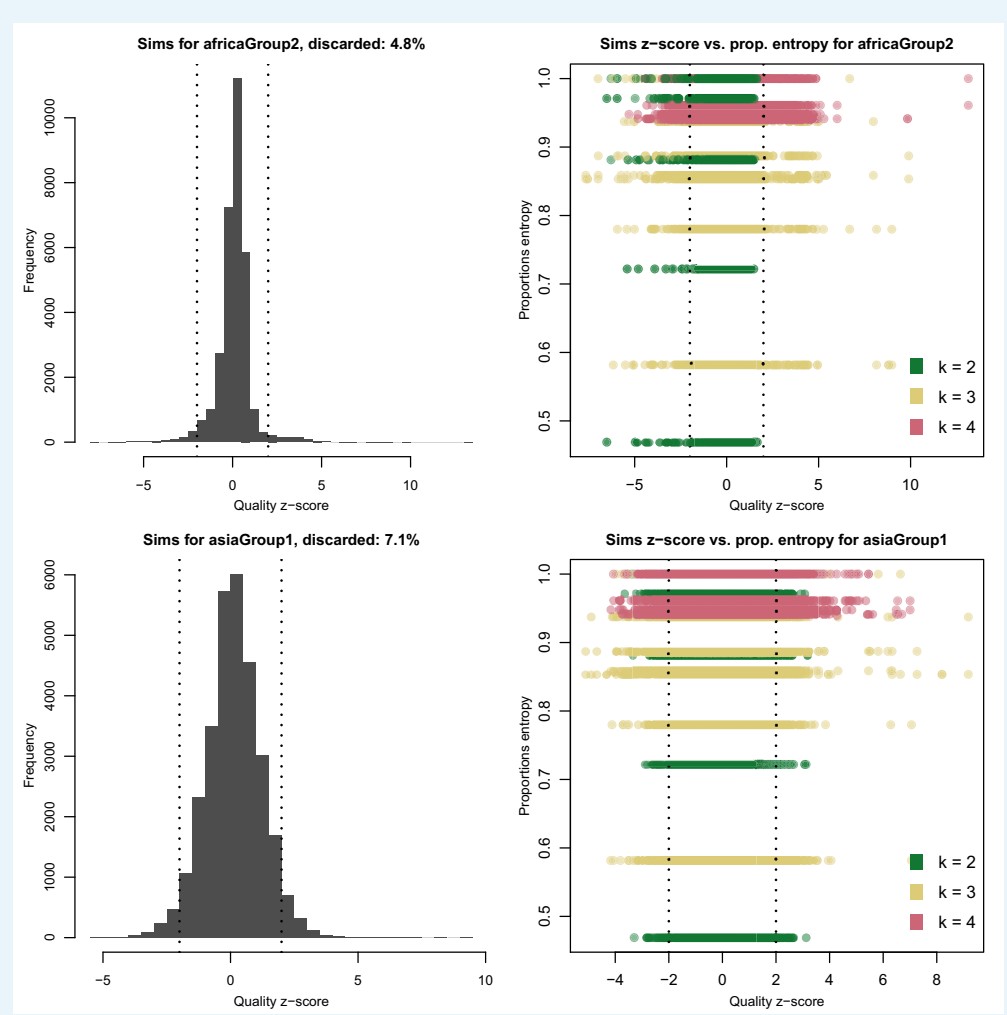

**Appendix 2—figure 1.** Distribution of quality scores haplotypes deconvolved from in silico mixtures using `DEploid`. Each row represents a different population (Africa and Asia). The left panels represent the overall distribution of z-scores whereas the right panels stratify results according to the entropy of mixture proportions (y-axis) and number of strains (color).

DOI: https://doi.org/10.7554/eLife.40845.025

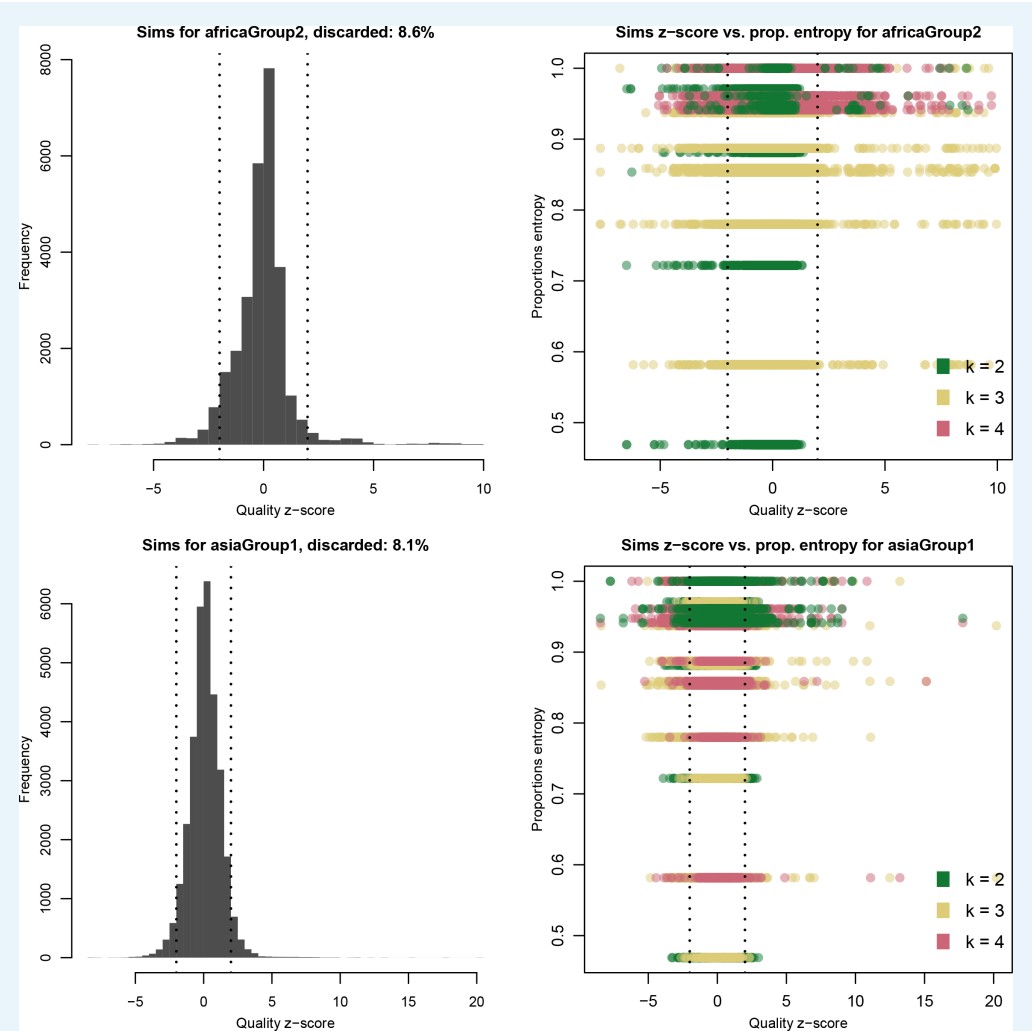

**Appendix 2—figure 2.** Distribution of quality scores haplotypes deconvolved from in silico mixtures using `DEploidIBD`. Each row represents a different population (Africa and Asia). The left panels represent the overall distribution of Z-scores whereas the right panels stratify results according to the entropy of mixture proportions (y-axis) and number of strains (color).
DOI: https://doi.org/10.7554/eLife.40845.026

## Appendix 3

DOI: https://doi.org/10.7554/eLife.40845.020

# Pf3k field sample analysis

## Sample choice

We used 2640 samples from the Pf3k project (see https://www.malariagen.net/projects/pf3k). We excluded Nigerian samples from downstream analysis as there were only five samples from this country. We discarded samples containing mixed malaria species and those where sequencing coverage depth was below 30X or in which less than 30% of sites were callable. Finally, we exclude all lab strains (including reference strains, artificial mixtures, and crosses samples) and duplicated samples. In total, we retained 2344 field samples from 13 countries.

# Data filtering

We ran `DEploid-IBD` on high quality biallelic SNPs (both coding and non-coding, tagged with PASS at the QUAL column in the VCF file) from Pf3k (**Pf3k Consortium, 2016**). Before the additional filtering step described below, this set contained 1,057,830 SNPs.

```
bcftools view\
--include FILTER=PASS" \
--min-alleles 2 \
--max-alleles 2 \
--types snps \
--output-file SNP_INDEL_Pf3D7_01_v3.high_quality_biallelic_snps.vcf.gz \
--output-type z \
SNP_INDEL_Pf3D7_01_v3.combined.filtered.vcf.gz
```

## High leverage data points

We found that markers with high coverage for both alleles could mislead our model, inducing it to fit additional strains. We used a threshold of >99.5% coverage (default) to identify markers with extremely high allele counts. We further expanded this list of potential problematic markers by considering their nearest 10 neighbours on both flanks, and excluding those that were tagged more than once (see *Appendix 3—figure 1*). These poorly-genotyped variants are likely to be errors of mapping and genotype calling.

To track down the causes of high leverage points, we assessed nucleotide diversity in the *P. falciparum* genome. We used clonal haplotypes to compute nucleotide diversity by running a sliding window along the genome. At each SNP, we use $n_0$ and $n_1$ to denote counts for reference and alternative alleles, respectively. Let $n = (n_0 + n_1)$ be the number of haplotypes in the population with a non-missing call. We computed the mean number of pairwise differences for this SNP as follows. First, we computed the total number of pairs as $n_{pairs} = n * (n - 1)/2$. Then, we computed the number of pairs that were the same, $n_{same} = (n_0 * (n_0 - 1)/2) + (n_1 * (n_1 - 1)/2)$, and the number of pairs that were different, $n_d = n_{pairs} - n_{same}$. Finally, we obtained the mean number of pairwise differences as $mpd = n_d/n_{pairs}$. To estimate nucleotide diversity $\pi$, we computed the sum of $mpd$ in a window of 20kbp centred on each SNP, and divided by the number of accessible bases, which produces the mean number of pairwise differences per base.

Regions containing high leverage points tended to be at the ends of chromosomes or within regions of high nucleotide diversity, where read mapping was problematic (see *Appendix 3—figure 2*). We identified potential outliers in all samples, and filtered out common outliers in at least 50 samples – 48,443 in total.

## Analysis preparation

To improve the accuracy and efficiency of the deconvolution process, we first split the data into groups based on genetic similarity. We computed genetic distance between two samples as follows:

$$d(x,y) = \sum_{l}^{L} WSAF_{x,l} * (1 - WSAF_{y,l}) + WSAF_{x,l} * (1 - WSAF_{y,l}) \qquad (7)$$

where $l$ represents an arbitrary locus, $L$ denotes the total number of loci, and $WSAF_{s,l}$ indicates the non-reference within-sample allele frequency for sample $s$ at locus $l$ is then given by $WSAF_{s,l} = \frac{a_{s,l}}{r_{s,l}+a_{s,l}}$ where $a_{s,l}$ is the number of read counts supporting the alternative allele in sample $s$ at locus $l$, and $r_{s,l}$ is the number of read counts supporting the reference allele in sample $s$ at locus $l$.

We found that samples from the same geographical region differentiated into clear groups. We used this initial grouping as the basis for defining the reference panels that assisted the deconvolution. The geographical groups arising from this analysis are listed below. In order to reduce computational time, we only used polymorphic sites at each population group:

1. Malawi, Congo, with 349,242 sites.
2. Ghana (Navrongo), with 508,606 sites.
3. Nigeria, Senegal, Mali, with 210,819 sites.
4. The Gambia, Guinea, Ghana (Kintampo), with 250,827 sites.
5. Cambodia (Pursat), Cambodia (Pailin), Thailand (Sisakhet), with 44,317 sites.
6. Vietnam, Laos, Cambodia (Ratanakiri), Cambodia (Preah Vihear), with 88,410 sites.
7. Bangladesh, Myanmar, Thailand (Mae Sot), Thailand (Ranong), with 84,868 sites.

## Haplotype quality assessment

In this work, we also assessed the quality of the haplotypes inferred by `DEploidIBD`. Our goal was to establish to what degree our inferred haplotypes were statistically indistinguishable, given a suite of population genetics statistics, from the subset of clonal haplotypes that had the same geographical origin. Our assumption was that haplotypes found in mixed infections would have similar characteristics than those present in clonal samples. In our assessment, we found that the distribution of statistics for groups of deconvoluted haplotypes had extreme outliers and presented a higher variance when compared with the clonal population originating on the same region. We noticed that the painting process implemented by `DEploid` struggles when faced with challenging mixtures. For instance, mixed infections in which the co-existing strains have the same relative proportion (e.g. $k = 4$ with each strain having a proportion of 25%), or samples in which proportions are very unbalanced (e.g. k = 2 with the marginal strain at 2%). This often results in an excess of alternative calls being assigned to one of the strains, which in turn provokes a deficit of diversity on the remaining haplotypes, that cannot be explained in terms of their genetic relationship to the reference genome used for mapping and assembly (3D7). We defined our quality metric as a $z$-score that approximates how much a deconvoluted haplotype deviates from the mean genetic diversity of the clonal population present in the same geographical area.

For each population, we computed the distribution of alternative calls observed within the subset of clonal samples ($k = 1$). Using this distribution as reference, we computed a $z$-score for each haplotype in the whole population following

$$z_i = \frac{a_i - \bar{a}_r}{\sigma_r},$$

where $a_i$ denotes the number of alternative calls in the haplotype $i$, and $\bar{a}_r$ and $\sigma_r$ are, respectively, the mean and standard deviation of observed alternative calls in the clonal set of samples from the population of origin. We only considere as suitable haplotypes with a $z$-score in the range $(-3, 3)$, thus discarding any strain that is three or more standard deviations away, in terms of alternative calls, from the mean observed for clonal samples. By using the set of

clonal samples as the reference distribution, we approximated the number of alternative calls expected in a genome belonging to that population, which serves as a proxy for genetic diversity but is easier to compute. Supp. *Appendix 3—figure 3* shows an example of this filtering process for the most problematic population in the dataset (Ghana). Supp. *Appendix 3—table 1* lists the number of haplotypes discarded by population while Supp. *Appendix 3—table 2* describes the number of haplotypes discarded by COI level. Statistical deconvolution of haplotypes in mixed infections remains a challenging problem and requires further research. Nonetheless, our quality metric can guide other researchers in the process of discarding haplotypes that are clearly artefactual

## Combining clonal sample pairs for background IBD computation

We combined randomly selected clonal sample pairs to create artificial mixed infections, as a way to generate a background IBD distributions for each country. We assumed these artificial mixed infections mimic infections generated from two independent mosquito bites. In this setting, strain proportions are determined by their median read depth, whereas sample coverage is obtained by accumulating the reference and alternative allele counts of two clonal samples. Similar to `DEploidIBD` deconvolution, SNPs with very high coverage resulting from this process caused high leverage in the model. Additionally, the sample sequence depth and skewness were heterogeneous due to different sample preparation and sequencing protocols. We reduced the `DEploidIBD` filtering threshold from 99.5% to 80%, and used low recombination probabilities to avoid false IBD breakpoint inference. We validated our method using lab crosses (*Miles et al., 2016*), and compared the IBD block detection using (*Li and Stephens, 2003*)'s painting with parental strains and the `DEploidIBD` algorithm (*Appendix 3—figure 4*).

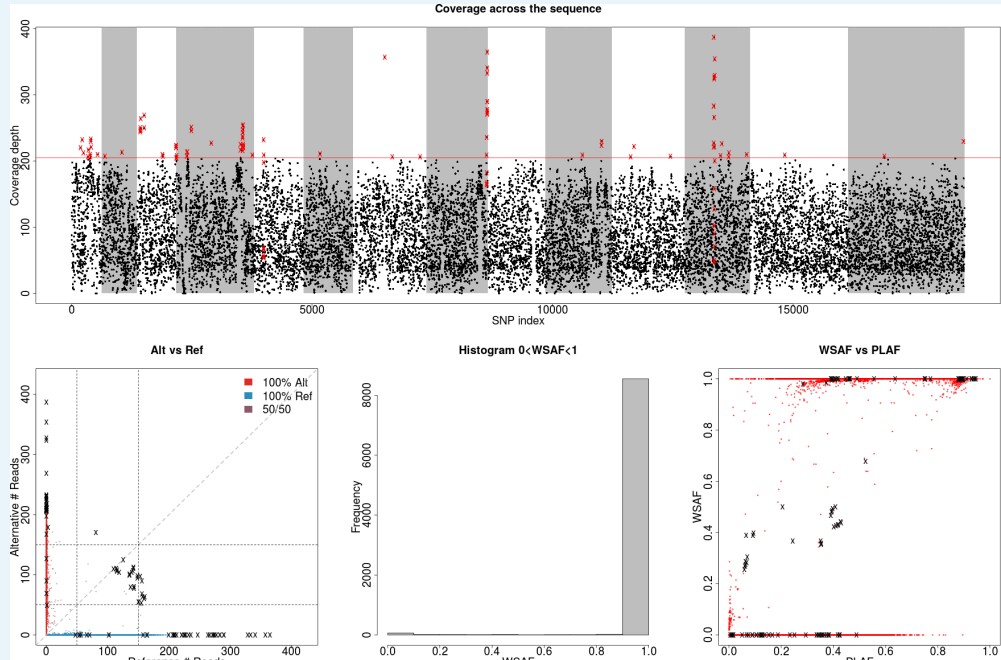

**Appendix 3—figure 1.** Identification of high leverage data points for filtering. (Top) Plot showing total allele counts across all markers for field isolate PG0415. We observe a small number of heterozygous sites with high coverage (shown as crosses on the bottom-left plot), which can potentially mislead our model to over-fit the data with additional strains (above the dotted line). We used a threshold of ≥99.5% coverage to identify markers with high allele counts. Red crosses indicate markers that are filtered out. (Bottom-left) Scatter plot showing

alternative against reference allele count. The marked black crosses refer to the outliers identified on the previous plot, which will cause the inference method to mistakenly identify the sample as being a mixed infection. (Bottom-middle) Histogram of allele frequency within sample. (Bottom-right) Allele frequency within sample (WSAF), compared against the population average (PLAF).

DOI: https://doi.org/10.7554/eLife.40845.028

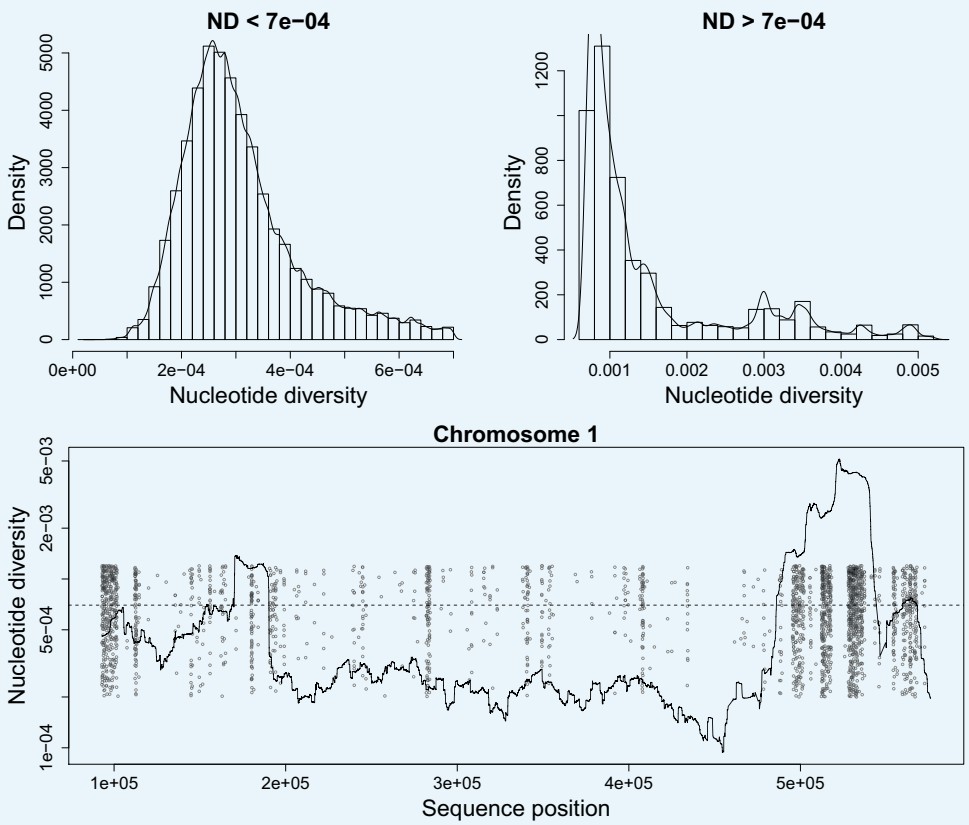

**Appendix 3—figure 2.** Nucleotide diversity for a sliding window size of 20,000 base pairs. (Top) Histograms showing the heavy tail of ND beyond 0.0007. (Bottom) Figure showing ND along *P. falciparum* chromosome 1. Scattered Points mark chromosome positions of poorly genotyped SNPs which we exclude from the deconvolution process. These points are jitterred to ease visualization.

DOI: https://doi.org/10.7554/eLife.40845.029

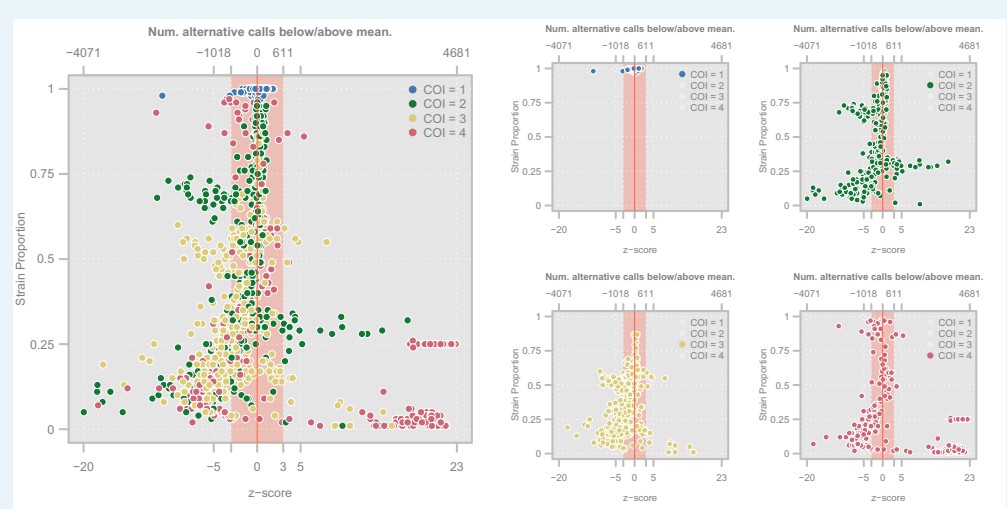

**Appendix 3—figure 3.** Diagnostic plots showing the distribution of haplotype quality ($z$-scores) for the Ghanian samples. Left. Scatterplot showing the relationship between haplotype $z$-score and strain proportion. The top axis shows the number of alternative calls below/above the mean of the subset of clonal samples that correspond to a given $z$-score. The vertical red line denotes a $z$-score of whereas the red-shaded area indicate the haplotypes we retain Point colors show the COI level of the sample. Right. Four views of the same plot in which the samples have been highlighted according to their COI level.
DOI: https://doi.org/10.7554/eLife.40845.030

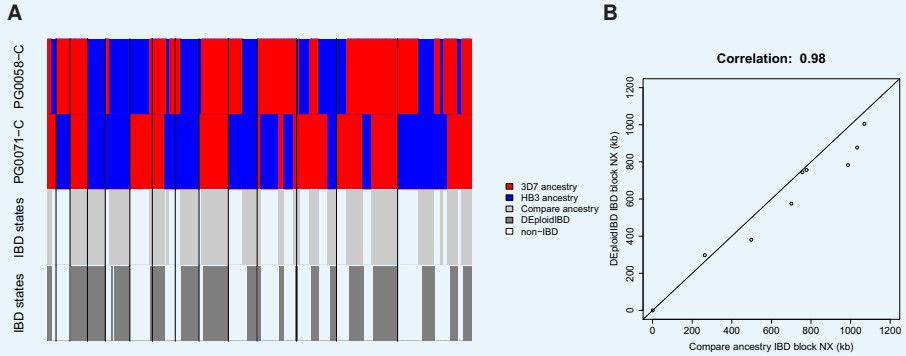

**Appendix 3—figure 4.** In silico validation of IBD estimation using lab crosses. (**A**) Visual summary of of IBD block detection between `DEploidIBD` (top) and ancestral state inference from *Li and Stephens (2003)* (bottom), using artificial mixtures of lab crosses PG0071-C and PG0058-C (last tract). (**B**) Scatter plot of IBD segment Nx values extracted by comparing clonal sample ancestry (using `DEploidIBD`) on artificial mixtures.
DOI: https://doi.org/10.7554/eLife.40845.031

**Appendix 3—table 1.** Number of haplotypes discarded and retained for each population in the Pf3k dataset.

| Country | Discarded | Retained | Fraction discarded |
|---|---|---|---|
| Bangladesh | 25 | 69 | 0.27 |
| Cambodia | 108 | 697 | 0.13 |
| DR. of Congo | 62 | 155 | 0.29 |
| Ghana | 493 | 609 | 0.45 |

*Appendix 3—table 1 continued on next page*

*Appendix 3—table 1 continued*

| Country | Discarded | Retained | Fraction discarded |
|---|---|---|---|
| Guinea | 79 | 88 | 0.47 |
| Laos | 28 | 110 | 0.20 |
| Malawi | 233 | 341 | 0.41 |
| Mali | 37 | 140 | 0.21 |
| Myanmar | 7 | 71 | 0.09 |
| Senegal | 2 | 167 | 0.01 |
| Thailand | 28 | 169 | 0.14 |
| The Gambia | 22 | 73 | 0.23 |
| Vietnam | 23 | 113 | 0.17 |
| Total | 1147 | 2802 | 0.29 |

DOI: https://doi.org/10.7554/eLife.40845.032

**Appendix 3—table 2.** Number of haplotypes retained and discarded stratified by COI level.

| COI | Retained | Discarded | Fraction discarded |
|---|---|---|---|
| 1 | 1331 | 34 | 0.02 |
| 2 | 669 | 291 | 0.30 |
| 3 | 583 | 533 | 0.48 |
| 4 | 219 | 289 | 0.57 |
| Total | 2802 | 1147 | |
| Fraction | 0.71 | 0.29 | |

DOI: https://doi.org/10.7554/eLife.40845.033

## Appendix 4

DOI: https://doi.org/10.7554/eLife.40845.020

# Expected levels of IBD in *P.falciparum* mixed infections

The amount of IBD observed in a mixed infection is a function of the number of oocysts present in the biting mosquito. We will demonstrate this below.

First, let us briefly review the fundamentals of malaria meiosis. In our case, we imagine a mosquito bites a human host containing two distinct malaria strains. Call these strains $A$ and $B$. Some number of gametocytes of $A$ and $B$ are imbided during the bite, differentiate into gametes, and undergo fertilization to produce zygotes (reviewed in *Ghosh et al., 2000*; *Bennink et al., 2016*). Some fraction of these zygotes succeed in establishing themselves as oocysts on the mosquito midgut (*Ghosh et al., 2000*). Three products of fertilization are possible, and thus the oocysts can be either: $A + A$ or $B + B$ (inbred oocysts, $n_{ii}$), or $A + B$ (outbred oocysts, $n_{ij}$). The oocyst state of a mosquito can be characterized by $(n_{ij}, n_{ii})$. Which strain is maternal and paternal may vary from oocyst to oocyst, but this is of no consequence here.

A $K = 2$ mixed infection is established when two distinct sporozoites, produced from the oocysts of this mosquito, infect a host. Each oocyst produces thousands of sporozoites (*Beier et al., 1991*), of four types (discussed in *McKenzie et al., 2001*), which pool in the mosquito salivary glands (*Ghosh et al., 2000*). Imagine drawing a $K = 2$ mixed infection from a mosquito harbouring a single outbred oocyst ($n_{ij}$=1). In such a mosquito there are two copies of each of the two strains (two sets of sister chromatids; $A$, $A$, $B$, $B$). Thus, ignoring recombination for the present, there are two pairs with an IBD fraction of 1 and, if our original strains are unrelated, the remainder of the $\binom{4}{2}$ pairs will have an IBD of 0. Thus a single $n_{ij}$ oocyst has an expected IBD of $E[\rho] = 2/\binom{4}{2} = 1/3$. We draw pairs without replacement because if sporozoites of only one type seed the infection, it will be $K = 1$. Importantly, neither recombination nor segregation change this result, as they only shuffle how the total identity is distributed between pairs, rather than create or destroy identity (identity is created by DNA replication and destroyed by mutation); the expectation is taken over all pairs and is thus unaffected.

Computing the expected IBD fraction for a mosquito possessing $n_{ij}$ outbred oocysts is an extension of the above. Again ignoring recombination, the expected IBD fraction $E[\rho|n_{ij}]$ is equal to the total number of pairs with an IBD of 1 (IBD pairs), over all possible pairs. In a mosquito with $n_{ij}$ oocysts, we have $2n_{ij}$ copies of each parental strain, thus we have $\binom{2n_{ij}}{2}$ IBD pairs for each parental strain, thus $2\binom{2n_{ij}}{2}$ IBD pairs total. Dividing this by the total number of pairs amongst $n_{ij}$ oocysts we have

$$E[\rho|n_{ij}>0] = \frac{2\binom{2n_{ij}}{2}}{\binom{4n_{ij}}{2}} = \frac{2n_{ij} - 1}{4n_{ij} - 1}. \tag{8}$$

The above yields $1/3$ for $n_{ij} = 1$, approaching $1/2$ as $n_{ij}$ grows. This result has been validated with `pf-meiosis` in *Appendix 4—figure 1*.

Including $n_{ii}$ oocysts is somewhat involved, as some pairs (selected without replacement) may be identical (thus yielding $K = 1$) or completely unrelated (yielding $K = 2$, but effectively without having undergone meiosis or producing any detectable recombination breakpoints between parental strains). We are interested in the expected IBD produced as a result of meiosis between parental strains, and thus for the moment we exclude these pairs. In practice, this means the mosquito must have at least one outbred oocyst, and at least one of the infecting sporozoites must be from an outbred oocyst.

The derivation is as above: first ignoring recombination and segregation, then enumerating all IBD pairs (pairs with IBD fraction of 1) and dividing by the total number of pairs to compute the expectation. Note that the additional IBD pairs possible between an outbred and inbred

oocyst are given by the term $8n_{ij}n_{ii}$, and that you can no longer use all possible pairs drawn without replacement as the denominator, but must exclude the pairs described above.

$$E[\rho|n_{ij}>0, n_{ii}] = \frac{2\binom{2n_{ij}}{2}+8n_{ij}n_{ii}}{2\binom{2n_{ij}}{2}+16n_{ij}n_{ii}+4n_{ij}^2}$$

$$= \frac{2(n_{ij}+2n_{ii})-1}{4(n_{ij}+2n_{ii})-1} \tag{9}$$

Which is of a similar form to above, but increases to $1/2$ quicker if more inbred oocysts are present. As before the equation is validated in **Appendix 4—figure 2A**.

The expression for $E[\rho|n_{ij}>0, n_{ii}]$ can also be derived by recognizing that there are three *types* of pairs possible in a mosquito with a collection of $n_{ij}$ and $n_{ii}$ oocysts: (1) a pair can contain two strains from a single $n_{ij}$, $(n_{ij}^{o=1})$; (2) a pair can contain two strains from two different $n_{ij}$, $(n_{ij}^{o=2})$; or (3) a pair can contain one strain from an $n_{ij}$ oocyst and one from an $n_{ii}$ oocyst, $n_{ij,ii}^{o=2}$. Pair type (1) is unique to malaria and has an $E[\rho|n_{ij}^{o=1}] = 1/3$, as shown above; pair type (2) are standard siblings with $E[\rho|n_{ij}^{o=2}] = 1/2$; and pair type (3) represent a mother-daughter relationship, also with $E[\rho|n_{ij,ii}^{o=2}] = 1/2$. The full IBD fraction and IBD segment length distributions of these pairs were generated using `pf-meiosis` and can be seen in **Appendix 4—figure 2B**. We enumerate the number of each pair type given $n_{ij}$ and $n_{ii}$, weighted by their expectation, to derive $E[\rho|n_{ij}>0, n_{ii}]$:

$$E[\rho|n_{ij}>0, n_{ii}] = \frac{n_{ij}^{o=1}E[f|n_{ij}^{o=1}]+n_{ij}^{o=2}E[f|n_{ij}^{o=2}]+n_{ij,ii}^{o=2}E[f|n_{ij,ii}^{o=2}]}{n_{ij}^{o=1}+n_{ij}^{o=2}+n_{ij,ii}^{o=2}}$$

$$= \frac{n_{ij}\binom{4}{2}1/3+16\binom{n_{ij}}{2}1/2+16n_{ij}n_{ii}1/2}{n_{ij}\binom{4}{2}+16\binom{n_{ij}}{2}+16n_{ij}n_{ii}} \tag{10}$$

$$= \frac{2(n_{ij}+2n_{ii})-1}{4(n_{ij}+2n_{ii})-1}$$

As above.

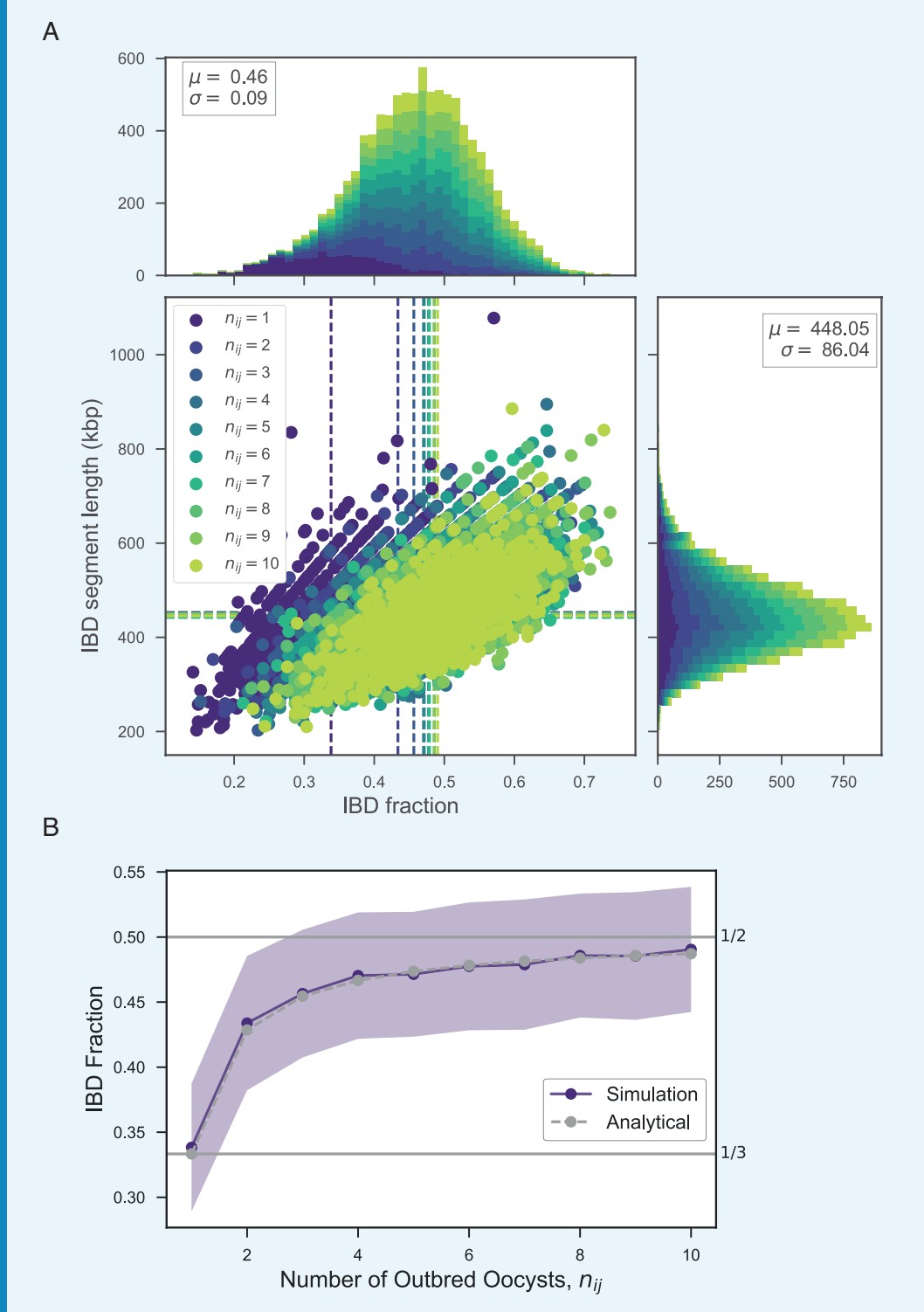

**Appendix 4—figure 1.** Exploring the relationship between number of outbred oocysts ($n_{ij}$) and IBD. (**A**) Joint IBD fraction and IBD segment length distributions for $K = 2$ mixed infections simulated from two unrelated strains and a fixed number of outbred oocysts $n_{ij}$, using `pf-meiosis`. Mean values for each distribution are indicated by same-color dashed lines. Each distribution is created from 1000 simulated mixed infections. (**B**) Validation of theoretical result given in text (S1.8). Line plot compares trend in expected IBD fraction with the number of outbred oocysts, $n_{ij}$, for infections simulated in panel A, and analytical expression S1.8.

DOI: https://doi.org/10.7554/eLife.40845.035

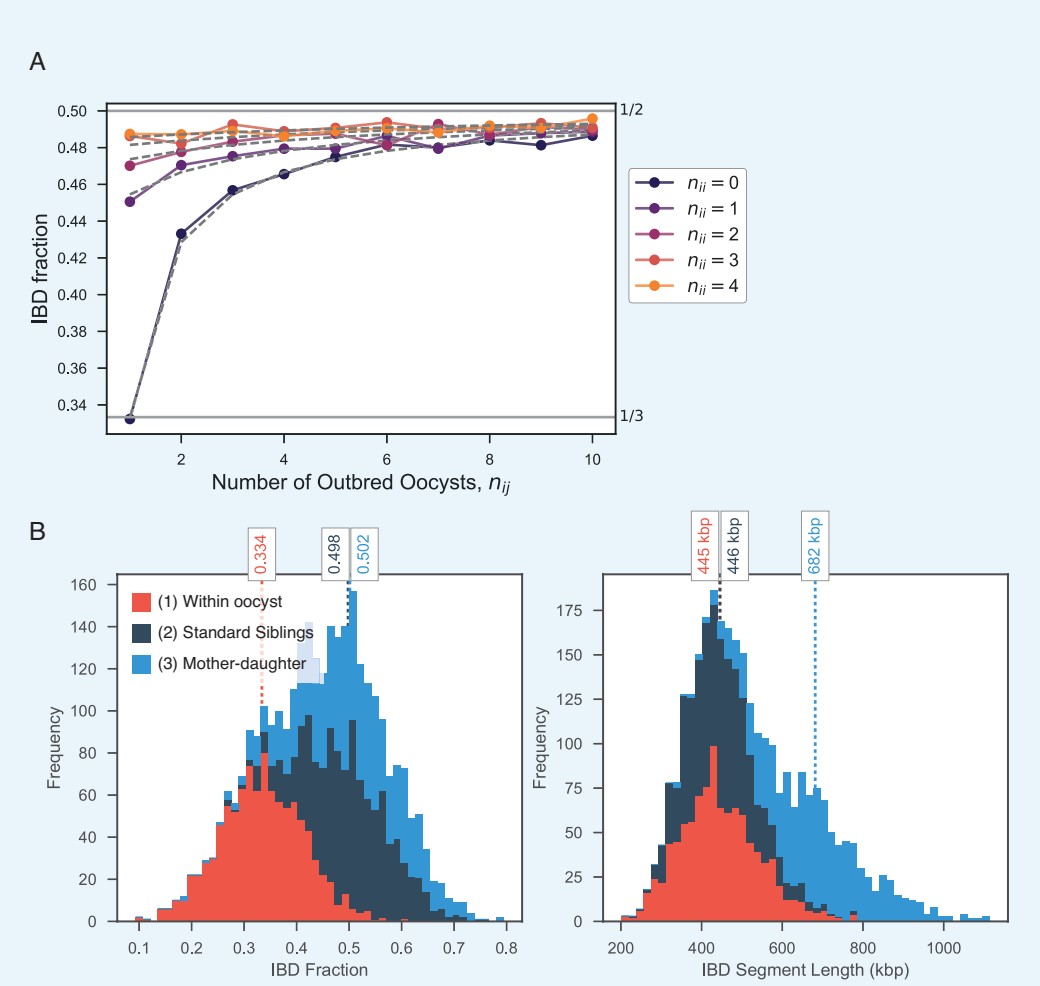

**Appendix 4—figure 2.** Exploring expected IBD allowing for outbred ($n_{ij}$) and inbred ($n_{ii}$) oocysts. (**A**) Validation of expression for expected IBD fraction conditional on outbred $n_{ij}$ and inbred $n_{ii}$ oocysts (S1.9). Line plot compares trend in expected IBD fraction with varying number of outbred (x-axis, $n_{ij}$) and inbred (line color, $n_{ii}$) oocysts and the analytical expression S1.9 (grey dashed lines). (**B**) Using pf-meiosis to simulate $K = 2$ mixed infections generated from (1) two strains from the same outbred oocyst from ($n_{ij}^{o=1}$, 'Within oocyst'); (2) two strains different outbred oocysts($n_{ij}^{o=2}$, 'Standard Siblings'); (3) one strain from an outbred and one strain from an inbred oocyst ($n_{ij,ii}^{o=2}$, 'Mother-daughter').

DOI: https://doi.org/10.7554/eLife.40845.036

