## [Decision Letter]

Thank you for submitting your article "The origins and relatedness structure of mixed infections vary with local prevalence of *P. falciparum* malaria" for consideration by *eLife*. Your article has been reviewed by three peer reviewers, and the evaluation has been overseen by a guest Reviewing Editor and Eduardo Franco as the Senior Editor. The following individuals involved in review of your submission have agreed to reveal their identity: Rachel Daniels (Reviewer #1); Bryan Greenhouse (Reviewer #2); Steve Schaffner (Reviewer #3).

The reviewers have discussed the reviews with one another and the Reviewing Editor has drafted this decision to help you prepare a revised submission.

In the discussion among the reviewers, the opinion emerged that this work was more appropriate as a Tools and Resources submission than as a Research Article. One reviewer commented that "the "epi" part is a bit of a stretch but the tool is the real (potential) advance. Regardless, I do think that for the tool to be published still requires additional work on validation and on the clarity of the methods and validation results."

Summary:

The authors address an important and challenging problem – phasing genotypes from deep sequence data obtained from mixed-strain falciparum infections. The manuscript is essentially a methods paper and offers a significant advance on the authors' prior work in this area (DEploid) in that it explicitly considers relatedness (IBD) between strains found within the same infection, primarily as a result of co-transmission. The new tool is validated in lab and in silico mixtures of parasite chromosomes. Following this validation of performance, DEploidIBD is applied to samples from the pf3K database and correlated to data on local prevalence from the Malaria Atlas Project.

The manuscript is overall well-written, with clear pseudocode and well-explained supplementary material; however, there are additional considerations that could perhaps improve the conclusions from the analysis and better delineate the advantages of this improvement over both the previous version and other reports

Essential revisions:

1) In general, I find it difficult to clearly understand how well the method is expected to perform based on the simulations. Most of the comparisons presented are between DEploidIBD and DEploid, and the authors do convincingly show an overall relative improvement. However based on the extent of the simulations and way the results are presented it is difficult for the reader to understand in what settings the method it is likely to perform well and where it is likely to fail. This is critical for the reader to understand the utility of the method. Regarding the extent of the simulations, the laboratory mixtures perform well but with only four strains in the population and optimal reference panels this is not a realistic test. The authors go on to perform in silico mixtures from Asia and Africa, but the former is presented for up to 3 strains while the latter (a more challenging dataset) is only presented for up to two strains. The method as presented is designed to work for up to four strains, and thus should be tested as such. It is quite possible that the information content for some (many) of these cases presents an identifiability problem even with a perfect method, however it would still be useful to provide the reader with information as to the limitations of the data/method. It may also be useful to evaluate the sensitivity of the method to the correctness of the reference panel, for lab mixtures and potentially also for in silico sims.

2) It is difficult for me to understand how well the deconvolution process worked on the simulations. This may in part be a result of my not fully understanding the statistics presented. For switch errors, what are the number of true switches for comparison? How are these errors defined (e.g. if the switch occurs 1 SNP in the wrong direction is it still considered an error?). How is genotype error calculated? Is this the percent of SNPs incorrectly phased within each deconvoluted haplotype? Does the denominator for this error include trivial cases (homozygous calls) or the only the more meaningful heterozygous calls in that sample? This is critical to understand, given the high proportion of "low confidence" haplotypes called on real samples.

3) Because phasing haplotypes is one of the primary goals of the software, evaluating haplotype quality from unknowns is of critical importance. The mechanics of the haplotype quality assessment is explained in the supplementary material, but the authors should explain why outliers in the proportion of alternative calls would be a good indicator of a problematic haplotype. Are these alternative calls with respect to the 3D7 reference genome, or to the reference panel used by the software? I presume the former but if the latter this should be made more clear. It would also be good to evaluate this method for "QC" of the outputs on simulations, in particular more complex simulations as discussed in my comment above.

4) Related to #3, in looking at haplotype quality in the real data (Figure 3A) it appears that more than half of polyclonal samples from Asia and ~75% of polyclonal samples have poor quality haplotypes. These numbers are not reflected well in Appendix 3—table 1, which lump in the (trivial) monoclonal samples, and Appendix 3—table 2 which lump together Asia and Africa. I also cannot reconcile the numbers in Appendix 3—table 2 with Figure 3A, e.g. >80% of COI==4 samples appear poor in Figure 3A but Appendix 3—table 2 mentions this figure at 57%. Perhaps the table and the figure representing different things, but if so I missed that distinction.

5) The impact of the population level correlations of output metrics with epidemiologic data are overstated in a few places in the manuscript, in particular in the impact statement which says that "monitoring fine scale patterns of mixed infections and within sample relatedness will be highly informative for assessing the impact of interventions." It is possible that with additional data and likely some sophisticated modeling this will be turn out to be true, but the general correlations with epidemiologic data presented in this manuscript, while qualitatively showing relationships in the expected direction (for Africa, but not in Asia) are neither unexpected nor quantitative in any way that would provide actionable information beyond basic surveillance data. The potential is certainly there, but the data presented do not support this claim.

6) The results in this manuscript, based on samples collected primarily in symptomatic individuals, cannot be generalized to infections in general, the vast majority of which are asymptomatic in moderate to high transmission areas, and in which the relative probability of superinfection to co-transmission is likely much greater since infections more easily stack up in semi-immune individuals with less chance of symptoms and thus treatment. E.g. "more than half of mixed infections arise from the transmission of siblings".

7) There are major concerns over validation and since this is a methods paper, that is a key component so that the reader can assess the usefulness in their own situation.

The method was validated by analyzing:

A) 27 experimental mixtures of 1 – 3 strains with no IBD betweenthem. The arrangement should provide an easy test for the method,since it employs highly divergent strains and an ideal referencepanel. DEploidIBD performs as well as DEploid, except for one casewhich it gets wrong. The authors comment that "LD information isnecessary to achieve accurate deconvolution". This confuses me, since as described both methods include LD information through the reference panel. What does this mean, and why did the new method perform more poorly than the original?

B) Simulated chromosomes with IBD made of mixtures of (a) pairs of

Asian strains, (b) pairs of African strains, (c) triplets of Asianstrains with a rather artificial IBD pattern. My concern here is thatthe performance of the method has not been tested over the relevantrange of complexity (e.g. that seen in this study). How does it workfor more realistic combinations of 3 strains, or for 3 Africanstrains, or for 4 strains? The method could still be quite valuableeven if it cannot handle something like 4 strains, but it would bevery helpful to know its limits. Also, why was (b) not compared to theperformance of DEploid?

8) The quality of reconstructed haplotypes is determined by comparison with monoclonal samples: haplotypes that look either too similar or too dissimilar to the reference are discarded. Are these haplotypes discarded anywhere besides the meiosis analysis? Are they included in COI estimates? More importantly, the number of discarded haplotypes is sometimes very large, e.g. 45% of all haplotypes for Ghana are discarded, and 57% for COI=4 from all populations. It is hard to have confidence in the remaining haplotypes when so many are clearly wrong, especially since there is no independent basis for exclusion. Further treatment of these poor quality haplotypes how they are generated or how they can be prevented, or what output is unaffected by them is needed.

[Editors' note: further revisions were requested prior to acceptance, as described below.]

Thank you for resubmitting your work entitled "The origins and relatedness structure of mixed infections vary with local prevalence of *P. falciparum* malaria" for further consideration at *eLife*. Your revised article has been favorably evaluated by Eduardo Franco as the Senior and Reviewing Editor, and three reviewers.

The manuscript has been improved. All reviewers acknowledged the substantial improvements and the extra work that was required. They also acknowledged the detailed and thoughtful responses you provided to address the first review round. However, there are some remaining concerns that they raised and which need to be addressed before acceptance, as outlined below.

Comments from the reviewers (edited and rearranged for concision and clarity):

1) It is still not clear how the authors differentiate the probabilities of identity-by-state from identity-by-descent, or how they account for this difference in their methodologies.

2) A few items remain unclear:

a) Appendix 1, subsection “Modelling mixed infections with IBD”: *p_rec_* is introduced here, but it is not clear how it is calculated.

b) Appendix 1, subsection “In silico mixtures of two field strains”: How are the regions of identity between genomes formed, i.e. how are breakpoints set between segments?

c) The handling of the number of strains (K) is not explained well. The authors say "The priors on the number of strains, K, and their proportions, w, are as in Zhu et al., 2018, with associated hyperparameters being user-defined", but that paper doesn't describe priors or associated hyperparameters for K. Rather, it says the number of strains is fixed at a high value and minor strains dropped in the end. This approach also seems to be reflected in the subsection “MCMC parameters for deconvolution”, of Appendix 1,. Since calculation of other parameters is conditional on K, I find this to be quite confusing.

3) The authors have done a great deal of additional simulation to more fully explore the effectiveness of their algorithm. My only suggestion here is to have a few lines summarizing their conclusions about the situations in which the program works well; ideally, this information should appear in the Abstract as well, since it is one of the more important take-aways from the paper.

4) More context and discussion are now provided about the pruning of haplotypes, which is good. There does seem to be curious gap, however. The error rate for reconstructed haplotypes is determined for simulated complex infections, and low-quality haplotypes are identified and removed from real complex infections, but I do not see any connection described between these two activities. Were low quality haplotypes filtered out from the simulated data? Were there any low-quality simulated haplotypes? The simulated dataset is the obvious context for assessing the effectiveness of the filtering procedure, so it's puzzling that this is not described.

5) Figure 2 attempts to convey a large amount of information, and does this very nicely. A few things still took me some time to decipher and wonder if there is any other way to make this easier for a reader. One was the horizontal histograms of K – the bins go from top to bottom and are indicated by value/α of the orange and blue but since there is a lot of gray and absent bins is took a while to figure this out despite the key at the top. I don't have a great suggestion as numerals would likely be too cluttering, but maybe at least mentioning explicitly in the figure legend that K=1 is at the top and K=4 at the bottom? Also, do the "coverage below 20" mean samples with median sequencing depth <20x? Maybe define explicitly in the legend?

6) Figure 3 re: per site genetic error, because many sites in a given sample will have homozygous calls and always be assigned correctly, it could be useful to convey what the (e.g. mean) null expectation would be as a benchmark, i.e. if assignment occurred at random. I'm not sure how easy this is to do or would be worth it, but would make the error result more interpretable (and likely might vary for different scenarios).

---

## [Author Response]

Summary:The authors address an important and challenging problem – phasing genotypes from deep sequence data obtained from mixed-strain falciparum infections. The manuscript is essentially a methods paper and offers a significant advance on the authors' prior work in this area (DEploid) in that it explicitly considers relatedness (IBD) between strains found within the same infection, primarily as a result of co-transmission. The new tool is validated in lab and in silico mixtures of parasite chromosomes. Following this validation of performance, DEploidIBD is applied to samples from the pf3K database and correlated to data on local prevalence from the Malaria Atlas Project.The manuscript is overall well-written, with clear pseudocode and well-explained supplementary material; however, there are additional considerations that could perhaps improve the conclusions from the analysis and better delineate the advantages of this improvement over both the previous version and other reports

We thank the editor and reviewers for their positive comments. We agree that this is essentially a methods paper, though the findings about the origins of mixed infection (i.e. c. 50% being through a single bite) and correlations with prevalence are both novel and biological.

Essential revisions:1) In general, I find it difficult to clearly understand how well the method is expected to perform based on the simulations. Most of the comparisons presented are between DEploidIBD and DEploid, and the authors do convincingly show an overall relative improvement. However based on the extent of the simulations and way the results are presented it is difficult for the reader to understand in what settings the method it is likely to perform well and where it is likely to fail. This is critical for the reader to understand the utility of the method. Regarding the extent of the simulations, the laboratory mixtures perform well but with only four strains in the population and optimal reference panels this is not a realistic test. The authors go on to perform in silico mixtures from Asia and Africa, but the former is presented for up to 3 strains while the latter (a more challenging dataset) is only presented for up to two strains. The method as presented is designed to work for up to four strains, and thus should be tested as such. It is quite possible that the information content for some (many) of these cases presents an identifiability problem even with a perfect method, however it would still be useful to provide the reader with information as to the limitations of the data/method. It may also be useful to evaluate the sensitivity of the method to the correctness of the reference panel, for lab mixtures and potentially also for in silico sims.

To address these comments we have carried out a much more comprehensive analysis of DEploidIBD, in order to identify scenarios under which the method should perform well or poorly. In particular, we have added a suite of in silicosimulation-based validation experiments for both African and Asian samples, trialling mixed infections of K=2, K=3, and K=4 with various proportions and IBD profiles (see the expanded ‘Method Validation’ section of the Results and Figure 2.). These include four distinct IBD profiles for the simulated K=2 infections (0%, 25%, 50% and 75% IBD) across 5 proportions, including the challenging equal-proportion case. For the K=3 infections, we simulated three IBD profiles designed to represent the IBD structure expected in instances where the infection was generated from one, two, or three independent bites; here across 7 proportion values, again including the equal-proportion case. Similarly, for K=4 infections, we simulated five IBD profiles, designed to capture the IBD structure expected in the case of one, two, three, or four bites generating the infection; across 5 proportions including the all-equal proportion case. In total this resulted in 122 in silico examples of mixed infections on which to trial the method.

Several observations emerged from this additional validation, such as:

- DEploidIBD performs better than DEploid in scenarios where IBD levels are high

- Equal-proportion cases are challenging for both DEploid and DEploidIBD

- Barring the equal-proportion cases, the K, proportion, and IBD statistics inferred from DEploidIBD seem to be largely accurate for K=2 and K=3 infections

- For K=4 cases, performance was poor for both DEpoid and DEploidIBD, although DEploidIBD performed marginally better.

These results are now all included in the revision and, combined, allow the reader a much better understanding of the strengths and limitations of the new method (specifically note the highly revised Figure 2s and 3).

2) It is difficult for me to understand how well the deconvolution process worked on the simulations. This may in part be a result of my not fully understanding the statistics presented. For switch errors, what are the number of true switches for comparison? How are these errors defined (e.g. if the switch occurs 1 SNP in the wrong direction is it still considered an error?). How is genotype error calculated? Is this the percent of SNPs incorrectly phased within each deconvoluted haplotype? Does the denominator for this error include trivial cases (homozygous calls) or the only the more meaningful heterozygous calls in that sample? This is critical to understand, given the high proportion of "low confidence" haplotypes called on real samples.

We acknowledge the rationale for using the statistics was not clear and have added a description for the error analysis model in Appendix 1 subsection “Error analysis”, including a link to the code used to calculate the statistics. The revised Figure 3 provides additional quantification of inferred haplotype quality under a much wider range of scenarios than previously. We now write:

“For haplotype quality analysis, we compared the inferred haplotype with the true haplotype. [...] An example analysis for three strains is shown in Figure 1. “

3) Because phasing haplotypes is one of the primary goals of the software, evaluating haplotype quality from unknowns is of critical importance. The mechanics of the haplotype quality assessment is explained in the supplementary material, but the authors should explain why outliers in the proportion of alternative calls would be a good indicator of a problematic haplotype. Are these alternative calls with respect to the 3D7 reference genome, or to the reference panel used by the software? I presume the former but if the latter this should be made more clear. It would also be good to evaluate this method for "QC" of the outputs on simulations, in particular more complex simulations as discussed in my comment above.

Extracting haplotypes from mixed infections was certainly a goal for DEploidIBD, though we do not see it, nor intended to present it, as being the primary one. Rather, both the primary goal and novelty of DEploidIBD was the extraction of IBD profiles from mixed infections, given their clear epidemiological significance. Obtaining high-quality haplotypes from complex mixed infections is a very difficult problem and we do not wish to present it as fully solved.

Our goal in analysing haplotype quality was to define a metric (z-score) that could be useful for ourselves and others to identify inferred haplotypes of comparable quality to those obtained from clonal strains. Our investigations show that several of the haplotypes inferred from low coverage samples, or where there are minor strains present, have unusually high or low numbers of variant calls, suggestive of unresolved issues (underestimating K or strain ‘dropout’ respectively).

It is important to note (and we have now highlighted this in the paper) that DEploidIBD infers K, proportions, and IBD profiles *before* and independently of the haplotypes it produces. We did not stress this point sufficiently in the original paper. Critically, the haplotypes are not used in any of our downstream epidemiological analysis. We have made amendments to the text to ensure we clearly acknowledge outstanding issues with haplotype quality, to emphasize their independence from K, proportion and IBD profile inference, and to highlight that they have not been used in downstream analyses. Appendix 3 provides additional substantial detail on these analyses.

4) Related to #3, in looking at haplotype quality in the real data (Figure 3A) it appears that more than half of polyclonal samples from Asia and ~75% of polyclonal samples have poor quality haplotypes. These numbers are not reflected well in Appendix 3—table 1, which lump in the (trivial) monoclonal samples, and Appendix 3—table 2 which lump together Asia and Africa. I also cannot reconcile the numbers in Appendix 3—table 2 with Figure 3A, e.g. >80% of COI==4 samples appear poor in Figure 3A but Appendix 3—table 2 mentions this figure at 57%. Perhaps the table and the figure representing different things, but if so I missed that distinction.

The reviewer was correct and we have generated a new table, Appendix 3—table 1 to clarify the breakdown of poor-quality haplotypes both geographically and across K=2, K=3 and K=4 infections. Regarding the consistency of Figure 3A and Appendix 3—table 2, we failed to highlight that the figure presents percentages on a by-sample basis – i.e. the percentage of samples with at least one bad quality haplotype. In contrast, the supplementary tables refer to individual strains (i.e. considering all haplotypes and disregarding their aggregation within samples). We have clarified this point in the main text.

5) The impact of the population level correlations of output metrics with epidemiologic data are overstated in a few places in the manuscript, in particular in the impact statement which says that "monitoring fine scale patterns of mixed infections and within sample relatedness will be highly informative for assessing the impact of interventions." It is possible that with additional data and likely some sophisticated modeling this will be turn out to be true, but the general correlations with epidemiologic data presented in this manuscript, while qualitatively showing relationships in the expected direction (for Africa, but not in Asia) are neither unexpected nor quantitative in any way that would provide actionable information beyond basic surveillance data. The potential is certainly there, but the data presented do not support this claim.

We acknowledge that our results raise more questions than they answer about how genetic data might be used to monitor interventions in real time, though we maintain that features such as the ability to quantify mixed infections and to infer the origins of such events are likely to be useful. In response to these criticisms we have completely rewritten the Discussion, providing a much more balanced view of what this work does and does not say about the value of genetic data in monitoring malaria epidemiology and highlight settings – such as in Asia – where observed patterns suggest that we need a much better understanding of how epidemiological fluctuations in space and time impact genetic diversity.

6) The results in this manuscript, based on samples collected primarily in symptomatic individuals, cannot be generalized to infections in general, the vast majority of which are asymptomatic in moderate to high transmission areas, and in which the relative probability of superinfection to co-transmission is likely much greater since infections more easily stack up in semi-immune individuals with less chance of symptoms and thus treatment. E.g. "more than half of mixed infections arise from the transmission of siblings".

This is a good point which we have now taken into consideration by qualifying this statement in all places it occurs. In the Abstract we now write “…47% of symptomatic dual infections contain sibling strains…”; in the Introduction “…more than half of symptomatic mixed infections arise from the transmission of siblings…”; and in the Discussion “…we found that 47% of dual infections within the Pf3k Project likely arose through co-transmission…”.

7) There are major concerns over validation and since this is a methods paper, that is a key component so that the reader can assess the usefulness in their own situation.The method was validated by analyzing:A) 27 experimental mixtures of 1 – 3 strains with no IBD betweenthem. The arrangement should provide an easy test for the method,since it employs highly divergent strains and an ideal referencepanel. DEploidIBD performs as well as DEploid, except for one casewhich it gets wrong. The authors comment that "LD information isnecessary to achieve accurate deconvolution". This confuses me, since as described both methods include LD information through the reference panel. What does this mean, and why did the new method perform more poorly than the original?

The new algorithm (DEploidIBD) deconvolutes mixed infections in a two stage process. In the first stage it estimates strain number, proportions and IBD profiles using a model that does *not* use a reference panel. In the second, it fixes strain number and proportions and then estimates haplotypes *with* a reference panel (sa in DEploid). Consequently, if strain number is incorrectly estimated, strain haploytpes will be too. This is what happens in the case of equally balanced strain mixtures where, for example, a mixture of three strains with proportions each of ⅓, is fitted as a mixture of two strains with proportions ⅓ and ⅔. We have clarified this within the text and rewritten the supplementary material to highlight this two-stage approach to deconvolution.

B) Simulated chromosomes with IBD made of mixtures of (a) pairs ofAsian strains, (b) pairs of African strains, (c) triplets of Asianstrains with a rather artificial IBD pattern. My concern here is thatthe performance of the method has not been tested over the relevantrange of complexity (e.g. that seen in this study). How does it workfor more realistic combinations of 3 strains, or for 3 Africanstrains, or for 4 strains? The method could still be quite valuableeven if it cannot handle something like 4 strains, but it would bevery helpful to know its limits. Also, why was (b) not compared to theperformance of DEploid?

As described in Essential Revision #1, we have augmented the in silico validation to including K=2, 3, 4 for both Asian and Africa across a range of IBD structures and proportions.

8) The quality of reconstructed haplotypes is determined by comparison with monoclonal samples: haplotypes that look either too similar or too dissimilar to the reference are discarded. Are these haplotypes discarded anywhere besides the meiosis analysis? Are they included in COI estimates? More importantly, the number of discarded haplotypes is sometimes very large, e.g. 45% of all haplotypes for Ghana are discarded, and 57% for COI=4 from all populations. It is hard to have confidence in the remaining haplotypes when so many are clearly wrong, especially since there is no independent basis for exclusion. Further treatment of these poor quality haplotypes how they are generated or how they can be prevented, or what output is unaffected by them is needed.

We have addressed this issue in a previous comment (#3). In brief, we highlight here that we discard haplotypes that have a very significant excess or deficit of genetic diversity when compared with the clonal population of samples from the same geographical region. We agree with the reviewer that other tests may be in order (but see response to #3) to assure the quality of the non-discarded haplotypes, however we did not elaborate on this as we have not used deconvoluted haplotypes in any analysis; we did not make this clear in the original text. Regarding the causes that biases the deconvolution of haplotypes, we have identify problematic cases in which proportions are very well balanced or extremely unbalanced, but this is still work in progress and needs further research.

[Editors' note: further revisions were requested prior to acceptance, as described below.]

Comments from the reviewers (edited and rearranged for concision and clarity):1) It is still not clear how the authors differentiate the probabilities of identity-by-state from identity-by-descent, or how they account for this difference in their methodologies.

As with all approaches to detecting IBD, the signal that the algorithms are looking for is long stretches of IBS that results from very recent common ancestry (in the last few generations), while short stretches may result from older common ancestor events and sporadic allele matches can happen by chance (or potentially through error). The parametric model described in the Materials and methods defines exactly how this is achieved and the specific parameter values (also described in the Materials and methods) tune the algorithm to look for particular types of signal. We have attempted to make the method as clear as possible, though would be happy to consider specific suggestions as to where our description could be improved further.

2) A few items remain unclear:a) Appendix 1, subsection “Modelling mixed infections with IBD”: p_rec_ is introduced here, but it is not clear how it is calculated.

Details on how the probability of recombination between two sites (*p**rec*__) is calculated are provided in the Implementation Details. We now direct the reader to these when *p**rec*__is introduced.

b) Appendix 1, subsection “In silico mixtures of two field strains”: How are the regions of identity between genomes formed, i.e. how are breakpoints set between segments?

The breakpoints are set at exactly 25%, 50% or 75% of the chromosome length creating one contiguous tract of IBD. This models scenarios wherein a given bivalent has undergone a single crossover event (all chromosomes must undergo *at least* one crossover event to ensure homolog pairing), and a chromosome involved in this crossover event was transmitted to the mixed infection, resulting in that chromosome having a single contiguous IBD tract. Given that the majority of chromosomes in *P. falciparum* have a map length of about 1 Morgan (see Figure 3B of the reference *Miles* et al.in Genome Research) this IBD structure should be frequently observed in mixed infections (albeit across a continuous spectrum of IBD percentages).

c) The handling of the number of strains (K) is not explained well. The authors say "The priors on the number of strains, K, and their proportions, w, are as in Zhu et al., 2018, with associated hyperparameters being user-defined", but that paper doesn't describe priors or associated hyperparameters for K. Rather, it says the number of strains is fixed at a high value and minor strains dropped in the end. This approach also seems to be reflected in the subsection “MCMC parameters for deconvolution”, of Appendix 1,. Since calculation of other parameters is conditional on K, I find this to be quite confusing.

In practice there is no prior on the number of strains as it is set at a fixed value, i.e. in the joint prior (Equation 2) P(K) = 1. This value, and the threshold for discarding strains (set at 0.01) is what we meant by hyperparameters. To clarify this, which have changed the text to:

“The number of strains, K, and their proportions w, are as in (Zhu, 2017): K is fixed at a user-defined value (here, K=4) and strains below a fixed proportion threshold (here, 0.01) are discarded.”

3) The authors have done a great deal of additional simulation to more fully explore the effectiveness of their algorithm. My only suggestion here is to have a few lines summarizing their conclusions about the situations in which the program works well; ideally, this information should appear in the Abstract as well, since it is one of the more important take-aways from the paper.

We have added to the first paragraph of the Discussion to cover this more thoroughly:

“Validation work using simulated mixed infections illustrated that DEploidIBD performs well on infections of two or three strains and across a wide-range of IBD levels. We note that limitations and technical difficulties remain, including deconvoluting infections with more than three strains, handling mixed infections with highly symmetrical or asymmetrical strain proportions (e.g. K=3 with strains at 33%, or K=2 with one strain at 2%), analysing data with multiple infecting species, coping with low-coverage data, and selecting appropriate reference panels from the growing reference resources.”

Though we recognize a similar overview would be useful in the abstract, we have chosen not to add one, as we are already exactly at the 150 word limit and would like to keep current sentences covering biological findings as well.

4) More context and discussion are now provided about the pruning of haplotypes, which is good. There does seem to be curious gap, however. The error rate for reconstructed haplotypes is determined for simulated complex infections, and low-quality haplotypes are identified and removed from real complex infections, but I do not see any connection described between these two activities. Were low quality haplotypes filtered out from the simulated data? Were there any low-quality simulated haplotypes? The simulated dataset is the obvious context for assessing the effectiveness of the filtering procedure, so it's puzzling that this is not described.

To address this issue, we ran our haplotype filtering strategy on all of the ~57K haplotypes produced for the in silicovalidation and we have now included these results in the Haplotype Quality Assessment section of the of the supplementary materials (see Appendix 2—figure 1 and Appendix 2—figure 2). The results are qualitatively consistent with what was observed in the *Pf3k* data, namely that balanced or highly asymmetric proportions (displayed as a low or high entropy of proportion in the figures) yield more discarded haplotypes, as do mixtures with four strains. However, the rate at which haplotypes are discarded is substantially lower for the in silicomixtures than the field mixtures, likely reflecting greater complexity in the latter. We note that, in silico mixtures deconvoluted by DEploid tend to have somewhat fewer haplotypes discarded than those deconvoluted by DEploidIBD (e.g. DEploid discards 4.6% of haplotypes from Africa, DEploidIBD discards 8.6%), which likely reflects the stronger prior on haplotypes used in DEploid.

5) Figure 2 attempts to convey a large amount of information, and does this very nicely. A few things still took me some time to decipher and wonder if there is any other way to make this easier for a reader. One was the horizontal histograms of K – the bins go from top to bottom and are indicated by value/α of the orange and blue but since there is a lot of gray and absent bins is took a while to figure this out despite the key at the top. I don't have a great suggestion as numerals would likely be too cluttering, but maybe at least mentioning explicitly in the figure legend that K=1 is at the top and K=4 at the bottom? Also, do the "coverage below 20" mean samples with median sequencing depth <20x? Maybe define explicitly in the legend?

We have taken your suggestions and added to the figure legend to clarify:

“From the left to the right, the panels show the strain proportion compositions, distribution of inferred K in a vertically-oriented histogram (top: K=1, bottom: K=4) and using both methods: DEploid in orange and DEploidIBD in blue, effective number of strains, pairwise relatedness and IBD N50 (the latter two only for DEploidIBD).”

Also:

“Grey points identify experiments of low coverage data (median sequencing depth <20)…”

6) Figure 3 re: per site genetic error, because many sites in a given sample will have homozygous calls and always be assigned correctly, it could be useful to convey what the (e.g. mean) null expectation would be as a benchmark, i.e. if assignment occurred at random. I'm not sure how easy this is to do or would be worth it, but would make the error result more interpretable (and likely might vary for different scenarios).

As the reviewer says, the absolute genotype error will depend on the fraction of sites that are homozygous. For this reason, we only calculated accuracy at sites that are heterozygous in the sample or the sample-specific reference panel. Consequently, the expected genotype error under the null is sample specific – but certainly much higher than the range reported (i.e. off the scale). Likewise, the switch error could be as high as the number of sample heterozygous sites (minus one), which is also off the scale. We have clarified the approach taken in the legend to the figure.